# BLOCK VERIFICATION ACCELERATES SPECULATIVE DECODING

**Ziteng Sun**[*]
Google Research
`zitengsun@`

**Uri Mendlovic**[*]
Google Research
`urimend@`

**Yaniv Leviathan**[*]
Google Research
`leviathan@`

**Asaf Aharoni**[*]
Google Research
`asafaharoni@`

**Jae Hun Ro**[*]
Google Research
`jaero@`

**Ahmad Beirami**[*]
Google Research
`beirami@`

**Ananda Theertha Suresh**[*]
Google Research
`theertha@`

## ABSTRACT

Speculative decoding is an effective method for lossless acceleration of large language models during inference. It uses a fast model to draft a block of tokens which are then verified in parallel by the target model, and provides a guarantee that the output is distributed identically to a sample from the target model. In prior works, draft verification is performed independently token-by-token. Surprisingly, we show that this approach is not optimal. We propose *Block Verification*, a simple draft verification algorithm that verifies the entire block jointly and provides additional wall-clock speedup. We prove that the proposed mechanism is optimal in the expected number of tokens produced each iteration and specifically is never worse than the standard token-level verification. Empirically, block verification provides modest but consistent wall-clock speedups over the standard token verification algorithm of 5%-8% in a range of tasks and datasets. Given that block verification does not increase code complexity, maintains the strong lossless guarantee of the standard speculative decoding verification algorithm, cannot deteriorate performance, and, in fact, consistently improves it, it can be used as a good default in speculative decoding implementations.

## 1 INTRODUCTION

Large language models (LLMs) (Chowdhery et al., 2022; Touvron et al., 2023; Achiam et al., 2023; Gemini Team et al., 2023) are often decoded through autoregressive sampling, where generating $k$ tokens requires $k$ costly serial evaluations of the model. To improve generation latency, Leviathan et al. (2022) proposed *speculative decoding*, which enables an LLM to generate several tokens concurrently. In each iteration, conditioned on the current decoded prefix, a guess of the next block of $\gamma$ tokens is made by a fast drafter (*e.g.,* a small model or a heuristic). Each of the resulting $\gamma + 1$ prefixes are then evaluated by the large target model in parallel. To guarantee that the final output follows the same distribution as that of the large model, some of the generated tokens are accepted while others are rejected. The accepted tokens[1] are then appended to the prefix, and the process repeats until generation ends. See Figure 1 and Algorithm 3.

In Leviathan et al. (2022), the drafts are verified through a sequence of token-level rejection steps. More specifically, given a prefix $c$, let $X_1, X_2, \ldots, X_\gamma$ be one sample block of length $\gamma$ from the draft model $\mathcal{M}_s$, where $\forall i \leq \gamma, X_i \sim \mathcal{M}_s(\cdot \mid c, X^{i-1})$. Using the conditional distributions under the target large model $\mathcal{M}_b$ returned by the parallel evaluation step $(\forall 0 \leq i \leq \gamma, \mathcal{M}_b(\cdot \mid c, X^i))$, the algorithm iterates over the draft tokens sequentially, and accepts each token $X_i$ with probability

$$\min\left\{1, \frac{\mathcal{M}_b(X_i \mid c, X^{i-1})}{\mathcal{M}_s(X_i \mid c, X^{i-1})}\right\},\tag{1}$$

---

[*]All emails `@google.com`.

[1]With an extra token sampled from either a residual distribution or the large model distribution.

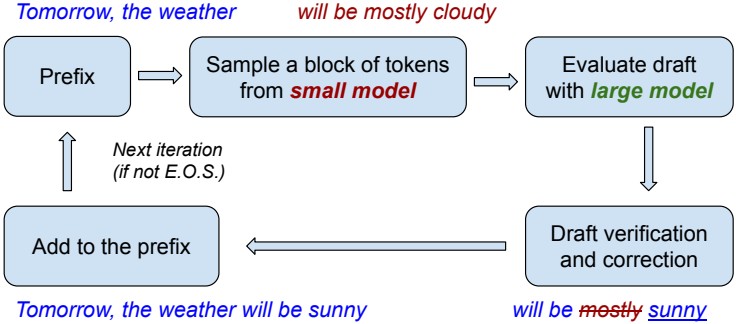

Figure 1: One iteration of speculative decoding (Algorithm 3). The prefixes and verified tokens are in blue, the unverified tokens from the draft model are in red, and the tokens sampled from the residual distribution are underlined.

The process continues until a token is rejected, at which point an extra token is sampled, for free, according to a residual distribution (see Algorithm 1 and Leviathan et al. (2022) for more details). We refer to this algorithm as *Token Verification*. Since its introduction in Leviathan et al. (2022), this token-by-token verification procedure has been the standard for follow-up works (see Section 7).

In this work, we make the surprising observation that *the standard token verification algorithm, is not optimal*, and propose a *strictly better method*. Our key observation is that we can increase the number of accepted tokens, while maintaining the identical distribution guarantee, by *jointly verifying the entire block of draft tokens* instead of verifying each token independently. Our proposed algorithm, which we call *Block Verification*, has the following advantages:

- **Simple to use.** The algorithm is a plug-and-play replacement of the standard token verification algorithm of speculative decoding. It does not incur additional computation or code complexity costs. See Algorithms 1 and 2 for a side-by-side comparison.
- **Identical distribution.** Importantly, our method is not an approximation and maintains the identical distribution guarantee of speculative decoding (Theorem 1).
- **Optimal improvement.** With the same drafting model, the speedup of block verification is no worse than that of standard token verification. Moreover, we show that block verification is an optimal verification procedure (Theorem 2).

We empirically test *block verification* and compare it with the standard *token verification* on a range of tasks and datasets. We show that our algorithm consistently improves over block efficiency (i.e. the expected number of generated tokens) by 7%-10% and overall empirical wall clock times by 5%-8% (see Table 1). Notably, our algorithm provides improvements only through the verification phase of speculative decoding, and hence the improvements can be combined with improvements obtained from other works that aim at improving the drafting phase. Since these improvements come for free, our block verification algorithm can be used as the draft verification algorithm by default in speculative decoding implementations.

## 2 A MOTIVATING EXAMPLE

The standard token verification algorithm stochastically rejects draft tokens with a higher probability from $\mathcal{M}_s$ than from $\mathcal{M}_b$. This is necessary to guarantee that the generated tokens follow the same distribution as that of $\mathcal{M}_b$. Our main observation is that considering whether to reject a block of draft tokens jointly, instead of one-by-one, can result in accepting more tokens. We now illustrate this through a simple example.

Consider the following trivial language model whose token space consists only of 2 tokens: A and B. Further, assume that both the large model $\mathcal{M}_b$ and the small model $\mathcal{M}_s$ are context-independent, and specifically that $\forall c$,

$$\mathcal{M}_b(\text{A}) = 1/3, \quad \mathcal{M}_b(\text{B}) = 2/3, \quad \mathcal{M}_s(\text{A}) = 2/3, \quad \mathcal{M}_s(\text{B}) = 1/3. \tag{2}$$

In this setting, token verification will accept each draft token $X$ independently with probability 1 if $X = \text{B}$ and $1/2$ if $X = \text{A}$. With a block size of $\gamma = 2$, since the total variation (TV) distance $d_{\text{TV}}(\mathcal{M}_b, \mathcal{M}_s) = 1/3$, the expected number of accepted tokens[2] from $\mathcal{M}_s$ with the token verification algorithm is $1 - 1/3 + (1 - 1/3)^2 = 10/9$ (see analysis in Leviathan et al. (2022)).

**An ideal algorithm with full information.** Suppose an algorithm can decide on what tokens to accept from $\mathcal{M}_s$ based on the *full joint distributions* of both tokens, *i.e.,*

$$\mathcal{M}_b(\text{AA}) = 1/9, \quad \mathcal{M}_b(\text{AB}) = 2/9, \quad \mathcal{M}_b(\text{BA}) = 2/9, \quad \mathcal{M}_b(\text{BB}) = 4/9,$$

$$\mathcal{M}_s(\text{AA}) = 4/9, \quad \mathcal{M}_s(\text{AB}) = 2/9, \quad \mathcal{M}_s(\text{BA}) = 2/9, \quad \mathcal{M}_s(\text{BB}) = 1/9.$$

The algorithm would have performed the following improved acceptance logic: always accept $X_1 X_2$ when $X_1 X_2 = \text{AB}$, BA, or BB since $\mathcal{M}_b(X_1 X_2) \geq \mathcal{M}_s(X_1 X_2)$, and accept AA with probability $\mathcal{M}_b(\text{AA})/\mathcal{M}_s(\text{AA}) = 1/4$ (correcting the samples to BB). The expected number of accepted tokens from $\mathcal{M}_s$ now becomes: $2(\mathcal{M}_s(\text{AB}) + \mathcal{M}_s(\text{BA}) + \mathcal{M}_s(\text{BB}) + 1/4 \times \mathcal{M}_s(\text{AA})) = 12/9 > 10/9$. This illustrates the benefit of considering the distribution of draft blocks jointly.

**Verification with partial information.** In general the full distribution over the next block of tokens is intractable to calculate. Instead, we only have access to the conditional distributions of the next token along the *sample path* of the draft block, $\mathcal{M}_b(\cdot \mid \boldsymbol{c}, X^i), \mathcal{M}_s(\cdot \mid \boldsymbol{c}, X^i)$ for various $i$'s. To emphasize, the ideal rejection logic does not need access to the full distribution, but care is needed in properly assigning the residual distribution. Our block verification does exactly this, as follows.

For the simple toy example describe above, we propose the following improved algorithm, which is a simplified version of the general block verification algorithm stated in Algorithm 2. When the draft tokens $X_1 X_2 = \text{AB}$ or BB, $\text{Pr}\,(\text{Accept}\,X_1 X_2) = 1$ similar to the idealized algorithm. When $X_1 X_2 = \text{AA}$, $\text{Pr}\,(\text{Accept}\,X_1 X_2) = 1/4$, and else the algorithm rejects both tokens and only corrects the first token to B since the algorithm doesn't have access to $\mathcal{M}_b(\cdot \mid \text{B})$. When $X_1 X_2 = \text{BA}$, it always accepts B, and then accepts A with probability $1/2$ (else it corrects the second token to B). Importantly, the marginal distributions of the generated tokens at the first token and the second token are always $\mathcal{M}_b(\cdot)$. Moreover, the algorithm only uses distributions that are conditioned on the *sample path* of the draft block, and hence it works in the partial information setting. We then simply add the generated tokens to the prefix and proceed to the next iteration. The expected number of accepted tokens is $2 \times (\mathcal{M}_s(\text{AB}) + \mathcal{M}_s(\text{BB})) + (1 + 1/2) \times \mathcal{M}_s(\text{BA}) + 1/4 \times 2 \times \mathcal{M}_s(\text{AA}) = 11/9$, which is better than the $10/9$ obtained by token verification. This example proves the following result:

**Lemma 1.** *The standard token verification algorithm of speculative decoding is not optimal.*

Note that while the expected number of accepted tokens in the example for block verification $(11/9)$ is higher than that of the standard token verification algorithm $(10/9)$, it is still less than that of the ideal algorithm with access to the full distribution $(12/9)$. In Section 4, we show that block verification is indeed optimal in the partial information case, with natural assumptions.

## 3 BLOCK VERIFICATION

In this section, we extend the above intuition to develop a general block verification algorithm, which works for standard speculative decoding with partial information. The high-level idea is to couple the acceptance of each draft token with other draft tokens. To do this, the algorithm considers draft sub-blocks with different lengths, and decides whether to accept each sub-block independently. The final accepted draft block is the longest accepted sub-block in the above process. The acceptance probabilities for each sub-block and the residual distributions are carefully chosen to maintain the distribution guarantee of the final output, and achieve optimal speedup.

See Algorithm 2 for a sketch implementation of block verification, and Algorithm 1 for a sketch implementation of the standard token verification for comparison. Note that the implementations follow the same overall structure (the differences are highlighted).

---

[2]This is different from the number of generated token in one iteration, which is the number of accepted tokens plus one (corrected token).

**Algorithm 1** Token Verification

**Input:** Draft block $X^\gamma$; small model distributions $\forall i < \gamma, \mathcal{M}_s(\cdot \mid \boldsymbol{c}, X^i)$; large model distributions $\forall i \le \gamma, \mathcal{M}_b(\cdot \mid \boldsymbol{c}, X^i)$.

1: Sample $\eta_1, \ldots, \eta_\gamma \sim U(0,1)$.
2: Set $\tau = 0$.
3: **for** $i = 1, \ldots \gamma$ **do**

4:     Set $h_i^{\text{token}} = \min\{\frac{\mathcal{M}_b(X_i \mid \boldsymbol{c}, X^{i-1})}{\mathcal{M}_s(X_i \mid \boldsymbol{c}, X^{i-1})}, 1\}$.
5:     **if** $\eta_i \le h_i^{\text{token}}$ **then**
6:       Set $\tau = i$.
7:     **else**
8:       **break.**
9:     **end if**
10: **end for**
11: **if** $\tau = \gamma$ **then**
12:    Sample $Y$ from $\mathcal{M}_b(\cdot \mid \boldsymbol{c}, X^\gamma)$.
13: **else**
14:    Sample $Y$ from $p_{\text{res}}^{\text{token}}(\cdot \mid \boldsymbol{c}, X^\tau)$ as in Equation (3).
15: **end if**
16: **Return** $X^\tau, Y$.

**Algorithm 2** Block Verification

**Input:** Draft block $X^\gamma$; small model distributions $\forall i < \gamma, \mathcal{M}_s(\cdot \mid \boldsymbol{c}, X^i)$; large model distributions $\forall i \le \gamma, \mathcal{M}_b(\cdot \mid \boldsymbol{c}, X^i)$.

1: Sample $\eta_1, \ldots, \eta_\gamma \sim U(0,1)$.
2: Set $\tau = 0$, $p_0 = 1$.
3: **for** $i = 1, \ldots \gamma$ **do**
4:     Set $p_i = \min\{p_{i-1} \frac{\mathcal{M}_b(X_i \mid \boldsymbol{c}, X^{i-1})}{\mathcal{M}_s(X_i \mid \boldsymbol{c}, X^{i-1})}, 1\}$.
5:     Set $h_i^{\text{block}}$ as in Equation (5).
6:     **if** $\eta_i \le h_i^{\text{block}}$ **then**
7:       Set $\tau = i$.
8:     **else**
9:       **continue.**
10:   **end if**
11: **end for**
12: **if** $\tau = \gamma$ **then**
13:    Sample $Y$ from $\mathcal{M}_b(\cdot \mid \boldsymbol{c}, X^\gamma)$.
14: **else**
15:    Sample $Y$ from $p_{\text{res}}^{\text{block}}(\cdot \mid \boldsymbol{c}, X^\tau)$ as in Equation (4).
16: **end if**
17: **Return** $X^\tau, Y$.

---

Residual distribution in Algorithm 1 (Line 15): $\forall x \in \mathcal{X}$,

$$p_{\text{res}}^{\text{token}}(x \mid \boldsymbol{c}, X^i) = \frac{\max\{\mathcal{M}_b(x \mid \boldsymbol{c}, X^i) - \mathcal{M}_s(x \mid \boldsymbol{c}, X^i), 0\}}{\sum_{x' \in \mathcal{X}} \mathcal{M}_b(x' \mid \boldsymbol{c}, X^i) - \mathcal{M}_s(x' \mid \boldsymbol{c}, X^i), 0\}}. \quad (3)$$

Residual distribution in Algorithm 2 (Line 15): $\forall x \in \mathcal{X}$,

$$p_{\text{res}}^{\text{block}}(x \mid \boldsymbol{c}, X^i) = \frac{\max\{p_i \cdot \mathcal{M}_b(x \mid \boldsymbol{c}, X^i) - \mathcal{M}_s(x \mid \boldsymbol{c}, X^i), 0\}}{\sum_{x' \in \mathcal{X}} \max\{p_i \cdot \mathcal{M}_b(x' \mid \boldsymbol{c}, X^i) - \mathcal{M}_s(x' \mid \boldsymbol{c}, X^i), 0\}}. \quad (4)$$

Acceptance probability in Algorithm 2 (Line 5): $h_\gamma^{\text{block}} = p_\gamma$, and when $i < \gamma$,

$$h_i^{\text{block}} = \frac{\sum_{x' \in \mathcal{X}} \max\{p_i \cdot \mathcal{M}_b(x' \mid \boldsymbol{c}, X^i) - \mathcal{M}_s(x' \mid \boldsymbol{c}, X^i), 0\}}{\sum_{x' \in \mathcal{X}} \max\{p_i \cdot \mathcal{M}_b(x' \mid \boldsymbol{c}, X^i) - \mathcal{M}_s(x' \mid \boldsymbol{c}, X^i), 0\} + 1 - p_i}. \quad (5)$$

Figure 2: The acceptance probabilities and residual distributions in Algorithms 1 and 2.

Importantly, token verification stops as soon as a token is rejected (the **break** in Line 9 of Algorithm 1), while block verification always operates on the full block. Equivalently, in token verification, $\tau = \arg\min\{\eta_i \le h_i^{\text{token}}\}$ while in block verification, $\tau = \arg\max\{\eta_i \le h_i^{\text{block}}\}$. This difference makes block verification tend to accept longer sub-blocks compared to token verification, resulting in higher block efficiencies. In Figure 3, we plot the empirical complementary CDF of the acceptance length for both algorithms with the toy models introduced in Equation (2) to demonstrate this.

See Algorithm 3 for the outer loop of the speculative decoding algorithm, which remains unchanged for both verification methods. See Figure 2 for additional definitions. See Appendix A for sketch Python implementations. Due to the simplicity of the change, the algorithm can be easily implemented without incurring additional code complexity in practical systems.

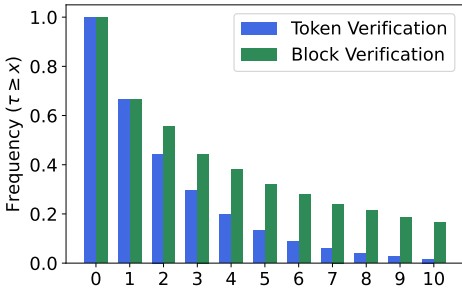

Figure 3: Empirical complementary CDF of $\tau$ for both algorithms with draft length $\gamma = 10$. The draft and target models are the context-independent toy models introduced in Equation (2).

---

**Algorithm 3** Speculative decoding (SPECDEC) (Leviathan et al., 2022)

**Input:** Prefix $c$, target model $\mathcal{M}_b$, draft model $\mathcal{M}_s$. Draft length $\gamma$. Verification algorithm VERIFY.

1: **while** E.O.S $\notin (X^\tau, Y)$ **do**
2:    Sample $X_1, \ldots, X_\gamma \sim \mathcal{M}_s(\cdot \mid c)$ using autoregressive sampling, keep the conditional
        probabilities at each step $\mathcal{M}_s(\cdot \mid c, X^i)$ for $i = 0, \ldots, \gamma - 1$.          {Obtain draft block.}
3:    Call the large model $\mathcal{M}_b$ and compute conditional probabilities $\mathcal{M}_b(\cdot \mid c, X^i)$
        for $i = 0, 1, \ldots, \gamma$ in parallel.                                              {Parallel scoring.}
4:    Get the accepted tokens with draft verification          {Draft verification and correction.}

$$X^\tau, Y = \text{VERIFY}(X^\gamma, \{\mathcal{M}_s(\cdot \mid c, X^i)\}_{i=0}^{\gamma-1}, \{\mathcal{M}_b(\cdot \mid c, X^i)\}_{i=0}^{\gamma}).$$

5:    $c \leftarrow c, X^\tau, Y$.                                              {Add decoded tokens to the prefix.}
6: **end while**

---

**Theoretical guarantees.** Speculative decoding with block verification preserves the distribution of its outputs (Theorem 1). Moreover, block verification achieves the *optimal* speedup among all valid draft verification algorithms in the outer loop of speculative decoding (Algorithm 3), resulting in a strict improvement over the standard token verification (Theorem 2). We defer the formal statements and the intuitions the parameter choices in Algorithm 2 to Section 4.

## 4   THEORETICAL GUARANTEES

In this section, we present the formal theoretical guarantees of block verification. Notably, that it produces the correct distribution and that it is optimal in terms of the expected number of generated tokens. Let $\mathcal{M}^*(\cdot \mid c)$ denote the distribution of the sequence up to the end of the generative process under model $\mathcal{M}$ and context $c$.

**Definition 1** (Valid draft verification algorithm). A draft verification algorithm VERIFY takes the draft block $X^\gamma$, small model distributions and large model distributions along the sample path, namely $\forall i < \gamma, \mathcal{M}_s(\cdot \mid c, X^i)$ and $\forall i \le \gamma, \mathcal{M}_b(\cdot \mid c, X^i)$ as inputs, and outputs a prefix $X^\tau, \tau \le \gamma$ of $X^\gamma$, and an additional token $Y$. VERIFY is said to be a valid draft verification algorithm if $\forall c$, models $\mathcal{M}_s, \mathcal{M}_b$, and block length $\gamma$, the outputs of Algorithm 3 (SPECDEC) with verification algorithm VERIFY satisfy

$$\text{SPECDEC}(c, \mathcal{M}_b, \mathcal{M}_s, \gamma, \text{VERIFY}) \sim_{\text{p}} \mathcal{M}_b^*(\cdot \mid c)^3, \tag{6}$$

*i.e.,* the distribution of the outputs is preserved.

Note for example that the standard token verification algorithm is a valid draft verification algorithm (Appendix A.1 in Leviathan et al. (2022)).

---

[3]We use $\sim_{\text{p}}$ to denote that two distributions are the same.

We now claim the following:

**Theorem 1.** *Block verification (Algorithm 2) is a valid draft verification algorithm.*

In other words, speculative decoding with block verification preserves the distribution of the output sequence.

We now further claim that block verification is optimal for all valid draft verification algorithms[4].

**Theorem 2.** *For $i > 0$, let $N(i)$ be the number of decoded tokens after $i$ iterations in Algorithm 3. For any valid draft verification algorithm $\textsc{Verify}$ in Definition 1, we have $\forall \boldsymbol{c}, \mathcal{M}_s, \mathcal{M}_b, \gamma$, and $i$,*

$$\mathbb{E}_{\textsc{BlockVerify}}[N(i)] \geq \mathbb{E}_{\textsc{Verify}}[N(i)],$$

*where the randomness is over the randomness of the draft block and the randomness of the algorithm.*

*In particular,*

$$\mathbb{E}_{\textsc{BlockVerify}}[N(i)] \geq \mathbb{E}_{\textsc{TokenVerify}}[N(i)].$$

In other words, among all valid verification algorithms, speculative decoding with block verification decodes the highest number of tokens in expectation in a fixed number of iterations. Note that since the computation overhead added by block verification is negligibly small, this establishes the overall optimality of the block verification algorithm. In particular, block verification provides a greater speedup than the standard token verification. We defer the proofs to Appendix B. Below we give intuitions on the algorithm changes that contribute to achieving the above guarantees.

**Intuition on parameter choices and theoretical guarantees.** The key quantities for achieving the speedup and distribution matching guarantees are $p_i$'s. In Lemma 3 in Appendix B.1, we show that $p_i$ corresponds to the probability that the sub-block $X^i$ will be kept in the final output. This is guaranteed by choosing the per-step acceptance probability properly since block verification keeps the longest accepted sub-block. Next we discuss how $p_i$'s contribute to the distribution matching and optimality guarantees.

*Distribution matching guarantee (Theorem 1).* To start, we ignore the minimum operation in the recursive definition of $p_i$'s. In such case, each $p_i$ is simply $\mathcal{M}_b(X^i \mid \boldsymbol{c})/\mathcal{M}_s(X^i \mid \boldsymbol{c})$, which is an upper bound on the actual $p_i$'s. As shown in Lemma 3, for any $X^i$, the probability that it is in the accepted block is $\mathcal{M}_s(X^i)p_i(X^i)$. Since the draft block $X^i$ is generated with probability $\mathcal{M}_s(X^i \mid \boldsymbol{c})$, this guarantees that the probability of getting $X^i$ by accepting it from the draft will be at most $\mathcal{M}_b(X^i \mid \boldsymbol{c})$, and hence the algorithm is not accepting $X^i$ more than needed.

The remaining part is to choose a suitable residual distribution $p_{\text{res}}^{\text{block}}$'s so that the distribution on the next token follows $\mathcal{M}_b(\cdot \mid X^i)$. Note that for any possible next token $x$, $(X^i, x)$ could also be obtained by accepting $X^{i+1}$ when $X_{i+1} = x$, with a probability of $\mathcal{M}_s(X^i, x)p_{i+1}(X^i, x)$, which should be subtracted to obtain the residual mass on $(X^i, x)$. This leads to the choice of $p_{\text{res}}^{\text{block}}$ in Equation (4) after proper normalization.

*Optimality guarantee (Theorem 2).* For optimality guarantee, the main proof is to show that for any prefix $X^i$ in the draft block, $p_i(X^i)$ is the maximum probability that a valid verification algorithm can accept $X^i$, which is stated in Lemma 4. This implies that in one iteration, block verification accepts the most tokens in expectation, and the multi-iteration case can be obtained by an induction argument.

To see that why $p_i(X^i)$ is the upper bound on the acceptance probability, we show that this is necessary to guarantee that for any prefix $X^i$ that could be obtained from multiple draft sample paths, the distribution over subsequent tokens are always the same $\mathcal{M}_b(\cdot \mid \boldsymbol{c}, X^i)$. This enables block verification to be used as a plug-and-play replacement of token verification in the outer loop of speculative decoding (Algorithm 3).

---

[4]We use $\textsc{BlockVerify}$ and $\textsc{TokenVerify}$ to denote block verification and token verification respectively when convenient.

Finally, we note that the optimality guarantee holds for all verification algorithms that can be used in Algorithm 3 as is. Specifically, there exist verification procedures that force the decoding logic to depend on the previous accept/reject decisions that produce more accepted tokens in average *in one iteration*. However, this will affect the decoding speed in subsequent iterations. In Appendix C, we present such an algorithm and name it *greedy block verification*. We empirically observe that block verification consistently outperforms it, so we include it mainly as a theoretical result.

## 5 EXPERIMENT SETUP

We conduct experiments using PALM-2 models (Chowdhery et al., 2022), and Vicuna models (Chiang et al., 2023).

For the experiments on PALM-2 models, we use PALM-2-S as the large target model and PALM-2-XXS / PALM-2-XXXS as the small drafter model. The order of the sizes of the models is PALM-2-XXXS < PALM-2-XXS < PALM-2-S. We evaluate on prompts from a wide range of datasets and tasks, including language modeling with one-billion language benchmark (LM1B) (Chelba et al., 2013), ChatGPT prompts sourced from LearnGPT (GPT Prompt) (Rashad, 2023), reasoning questions (WebQA) (Berant et al., 2013), physical commonsense reasoning questions (PIQA) (Bisk et al., 2020), scraped conversations with ChatGPT (ShareGPT) (Rashad, 2023; RyokoAI, 2023), summarization tasks (XSum) (Narayan et al., 2018), grade school math problems (GSM8K) (Cobbe et al., 2021), and German to English translation (WMT DeEn) (Bojar et al., 2014). For all datasets, we decode the first 1000 prompts using a max input prompt length of 512 and decode up to 128 output tokens. We use a batch size of 1 in all experiments except for the experiments in Appendix D.1. Note that since our method only modifies the verification phase of the algorithm and doesn't introduce additional draft tokens, the speedup we get is independent of the batch size. We use a temperature of 1.0 for the experiments on PALM-2 models.

For Vicuna family of models (Chiang et al., 2023), we conduct the set of experiments in Spec-Bench (Xia et al., 2024). We discussed detailed settings for these experiements in Section 6.2.

## 6 RESULTS

We focus our main experiments on the comparison between block verification and token verification. Recent works (Sun et al., 2023; Miao et al., 2023) have extended speculative decoding to the case with multiple draft blocks to improve block efficiency. However, these methods also increase the required computation from the large model to verify the drafts, which is undesirable when query batching is performed. We empirically show that our method outperforms these methods in the large batch setting even with only one draft block. We defer the results to Appendix D.1 and focus on the one draft case in the main section below.

### 6.1 EXPERIMENTAL RESULTS ON PALM-2 MODELS

We empirically compare speculative decoding with block verification to speculative decoding with token verification, and find that block verification provides small yet consistent improvements in a wide range of settings, both when measuring idealized *block efficiency* and real world *wall clock time*.

*Block efficiency* measures the speedup in an idealized settings where we neglect the evaluation time of the draft model and assume that we have enough compute capacity for evaluating the large model on all draft prefixes in parallel. Specifically, it measures the average number of decoded tokens per serial call to the target model. We observe consistent improvements for all datasets and draft models. For $\gamma = 8$ with PALM-2-XXS as the drafter, the improvement in block efficiency ranges from $7.00\%$ to $10.06\%$ with an average of $8.30\%$.

We also observe consistent improvements in *wall clock time*, which measures the actual speedup, including all the real-world overheads. See (Leviathan et al., 2022; Chen et al., 2023a) for a more detailed discussion of these overheads. For $\gamma = 8$ with PALM-2-XXS as the drafter, the improvement in block efficiency ranges from $5.36\%$ to $8.14\%$ with an average of $6.49\%$. The detailed numbers for this setting are listed in Table 1.

Table 1: Speedup comparison between token verification (TOKENV) and block verification (BLOCKV) with $\gamma = 8$ with PALM-2-S as the target model and PALM-2-XXS as the draft model on various datasets and tasks. We list the average and standard deviation across 3 runs with different seeds on 1000 test prompts.

| Dataset | Block efficiency | | | Wall clock time speedup over baseline | | |
|---|---|---|---|---|---|---|
| | TOKENV | BLOCKV | Improve. ↑ % | TOKENV | BLOCKV | Improve. ↑ % |
| LM1B | $3.21 \pm 0.01$ | $3.49 \pm 0.02$ | $8.68 \pm 0.79$ | $2.17 \pm 0.01$ | $2.32 \pm 0.01$ | $6.85 \pm 0.74$ |
| GPT Prompt | $3.41 \pm 0.04$ | $3.76 \pm 0.02$ | $10.06 \pm 1.66$ | $2.30 \pm 0.02$ | $2.48 \pm 0.01$ | $8.14 \pm 1.55$ |
| WebQA | $3.44 \pm 0.01$ | $3.70 \pm 0.01$ | $7.53 \pm 0.24$ | $2.32 \pm 0.00$ | $2.45 \pm 0.01$ | $5.75 \pm 0.22$ |
| PIQA | $3.40 \pm 0.02$ | $3.68 \pm 0.00$ | $8.30 \pm 0.62$ | $2.29 \pm 0.01$ | $2.44 \pm 0.00$ | $6.52 \pm 0.58$ |
| ShareGPT | $3.34 \pm 0.01$ | $3.62 \pm 0.03$ | $8.45 \pm 0.98$ | $2.25 \pm 0.01$ | $2.40 \pm 0.02$ | $6.68 \pm 0.91$ |
| XSum | $3.49 \pm 0.02$ | $3.76 \pm 0.01$ | $7.63 \pm 0.94$ | $2.35 \pm 0.01$ | $2.49 \pm 0.01$ | $5.82 \pm 0.88$ |
| GSM8K | $3.81 \pm 0.01$ | $4.15 \pm 0.03$ | $8.74 \pm 0.56$ | $2.55 \pm 0.01$ | $2.73 \pm 0.02$ | $6.84 \pm 0.51$ |
| WMT-DeEn | $3.19 \pm 0.01$ | $3.41 \pm 0.02$ | $7.00 \pm 0.78$ | $2.15 \pm 0.01$ | $2.27 \pm 0.01$ | $5.36 \pm 0.73$ |
| Average | 3.41 | 3.70 | 8.30 | 2.30 | 2.45 | 6.49 |

**The effect of draft length $\gamma$.** We also perform comparisons of the algorithms for other block lengths ($\gamma = 4$ and $\gamma = 6$) and observe consistent improvements. We plot the average improvement over all datasets in Figure 5 with the numbers in Figure 4. With the same drafter, the relative improvement of block verification over token verification increases as $\gamma$ increases. This is consistent with our intuition since when $\gamma = 1$, the two algorithms are the same and as $\gamma$ increases, block verification would benefit more from coordinating the acceptance rule considering the realization of all tokens in the draft block.

As shown in Figure 4, similar to token verification, the block efficiency of block verification increases as $\gamma$ increases. However, the wall clock speedup peaks at a certain draft length ($\gamma = 4$ or $\gamma = 6$ for all settings) due to the increased computation cost in the verification phase. Hence we focus on $\gamma \leq 8$ in our experiments.

**The effect of the drafter.** We also consider the effect of the quality of the drafter on the improvement. In Figure 4, we list the average block efficiency and wall clock speed up under different draft lengths for both drafters. Note that PALM-2-XXS is a larger model than PALM-2-XXXS, and hence a better drafter in terms of quality, as demonstrated by the better average block efficiencies in the table. In Figure 5, we plot the average improvement under different drafter models, PALM-2-XXS and PALM-2-XXXS. The improvements hold for both drafters. And the relative improvement in block efficiency under PALM-2-XXS is greater than that under PALM-2-XXXS. This shows that the improvement obtained from block verification can be combined with the improvement on the quality of the drafter, and the improvement might be more significant under better drafters.

Detailed results for experiments performed with different drafters, different datasets, and different draft lengths are listed in Appendix D.

## 6.2 EXPERIMENTAL RESULTS ON SPEC-BENCH WITH VICUNA MODELS

We also conduct the set of experiments proposed in Spec-Bench (Xia et al., 2024) with Vicuna family of models (Chiang et al., 2023). The benchmark includes various generation subtasks including multi-turn conversation, retrieval-augmented generation, summarization, translation, question answering, and mathematical reasoning. See Xia et al. (2024) for a detailed discussion of the subtasks. For all experiments in this section, we use a single NVIDIA H100 GPU with a batch size of 1 and a max generation length of 1024. We use Vincuna-7B-v1.3 as the target model and Vincuna-68M as the draft model. To study the effect of temperature, we consider temperatures in $\{0.2, 0.6, 1.0\}$ and fix $\gamma = 8$. The results are listed in Table 2. The reported numbers are the average of 3 runs.

| $\gamma$ | Drafter | TOKENV | | BLOCKV | |
|---|---|---|---|---|---|
| | | BE | WS | BE | WS |
| 4 | XXS | 2.89 | 2.44 | 2.99 | 2.50 |
| | XXXS | 2.35 | 2.36 | 2.43 | 2.43 |
| 6 | XXS | 3.23 | 2.43 | 3.43 | **2.54** |
| | XXXS | 2.50 | 2.39 | 2.63 | 2.50 |
| 8 | XXS | 3.41 | 2.30 | **3.70** | 2.45 |
| | XXXS | 2.57 | 2.28 | 2.73 | 2.40 |

Figure 4: Table on average block efficiency (BE) and wall clock speedup (WS) across all datasets for token verification (TOKENV) and block verification (BLOCKV) with different $\gamma$. The large model is PALM-2-S and the drafter model is either PALM-2-XXS (XXS) or PALM-2-XXXS (XXXS).

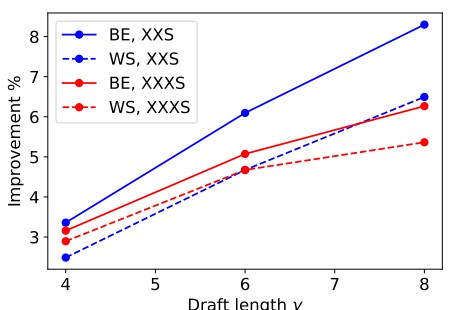

Figure 5: Average relative improvement of block verification over token verification in block efficiency (BE) and wall clock speedup (WS) across all datasets for different drafters and draft lengths.

Our algorithm obtains consistent improvement compared to token verification (up to 8.7% in block efficiency and up to 6.7% in wall clock speedup) across different draft lengths for all temperatures bigger than 0. This demonstrates the applicability of our method for different families of models.

**The effect of temperature.** Note that for temperature of 0, which corresponds to greedy decoding, our algorithm degenerates to token verification and doesn't provide additional speedups. In non-zero temperature settings, the advantage is consistent and the additional improvement is higher for larger temperatures. The observation is consistent with the intuition behind the algorithm, which obtains improvement on block efficiency by coordinating the randomness in the acceptance decisions at different token locations.

Table 2: Speedup comparisons between token verification (TOKENV) and block verification (BLOCKV) on Spec-Bench (Xia et al., 2024) for temperature $T \in \{0.2, 0.6, 1.0\}$. We use Vicuna-7B-v1.3 as the target model and Vicuna-68M as the draft model. $\gamma = 8$ for all experiments and each number is an average of 3 runs.

| $T$ | Block efficiency | | | Wall clock speedup over baseline | | |
|---|---|---|---|---|---|---|
| | TOKENV | BLOCKV | Improve. $\uparrow$ % | TOKENV | BLOCKV | Improve. $\uparrow$ % |
| 0.2 | 2.75 | 2.85 | 3.72 | 1.22 | 1.24 | 1.66 |
| 0.6 | 2.75 | 2.90 | 5.32 | 1.23 | 1.29 | 4.24 |
| 1.0 | 2.79 | 3.04 | 8.70 | 1.27 | 1.34 | 6.07 |

# 7 RELATED WORK

**Parallel decoding.** Our work improves speculative decoding (Leviathan et al., 2022), a framework for decoding several tokens concurrently. *Draft and verify* (Stern et al., 2018) was an earlier work, which proposed to independently predict and decode several tokens in parallel, for the greedy decoding case (zero temperature). Speculative decoding has later also been proposed in Chen et al. (2023a).

**Single draft improvements.** There have been many works aiming to improve speculative decoding without making use of more than one decoding draft. In Table 3, we list a set of works in the draft and verify framework with a breakdown of their drafting and verification algorithms. See Xia et al. (2024) for a comprehensive study. In the single-draft case, several works have worked

Table 3: Recent works based on the draft and verify framework. Temperature 0 refers to greedy decoding and non-zero temperature refers to sampling.

| Work | # drafts | Temp. | Drafting | Verification |
|------|----------|-------|----------|--------------|
| Stern et al. (2018) | 1 | 0 | parallel softmax layers | token matching |
| Yang et al. (2023) | 1 | 0 | additional text | token matching |
| Leviathan et al. (2022) | 1 | $\geq 0$ | small LM | TOKENVERIFY (Algorithm 1) |
| Chen et al. (2023a) | 1 | $\geq 0$ | small LM | TOKENVERIFY (Algorithm 1) |
| He et al. (2023) | 1 | $\geq 0$ | database retrieval | TOKENVERIFY (Algorithm 1) |
| Chen et al. (2023b) | 1 | $\geq 0$ | cascade of small LMs | TOKENVERIFY (Algorithm 1) |
| Sun et al. (2024) | 1 | $\geq 0$ | hierarchical drafters | TOKENVERIFY (Algorithm 1) |
| Zhou et al. (2023) | 1 | $\geq 0$ | distilled small LMs | TOKENVERIFY (Algorithm 1) |
| Liu et al. (2023) | 1 | $\geq 0$ | distilled small LMs | TOKENVERIFY (Algorithm 1) |
| Gloeckle et al. (2024) | 1 | $\geq 0$ | parallel softmax layers | TOKENVERIFY (Algorithm 1) |
| Zhang et al. (2024) | 1 | $\geq 0$ | layer skip | TOKENVERIFY (Algorithm 1) |
| Elhoushi et al. (2024) | 1 | $\geq 0$ | early exit | TOKENVERIFY (Algorithm 1) |
| **This work** | 1 | $\geq 0$ | small LM | BLOCKVERIFY (Algorithm 2) |
| Sun et al. (2023) | $\geq 2$ | $\geq 0$ | small LM | SpecTr |
| Miao et al. (2023) | $\geq 2$ | $\geq 0$ | small LM | multi-round TOKENVERIFY |
| Li et al. (2024) | $\geq 2$ | $\geq 0$ | small LM | multi-round TOKENVERIFY |
| Chen et al. (2024) | $\geq 2$ | $\geq 0$ | small LM | multi-round TOKENVERIFY |

on improving the drafting phase of speculative decoding (He et al., 2023; Chen et al., 2023b; Sun et al., 2024; Zhou et al., 2023; Liu et al., 2023; Gloeckle et al., 2024; Zhang et al., 2024; Elhoushi et al., 2024). However, these algorithms all use the same token verification algorithm. Our proposed block verification algorithm can be used in tandem with the drafting techniques in Table 3, yielding combined gains. We leave a more systematic study of the improvement of block verification in these cases for future study.

The only other work that we are aware of that improves the verification step in speculative decoding is the independent work of Hu and Huang (2024), which uses tree Monte Carlo to improve speculative decoding in the single draft case, and have proved that their algorithm improves over token verification. On the contrary, we prove that our algorithm achieves the optimal speedup among all valid verification algorithms, including theirs. Our algorithm also requires minimal changes to the original token verification algorithm, making it easy to implement and adapt everywhere in practice.

**Multiple drafts.** Recently, speculative decoding is extended to multiple drafts (Sun et al., 2023; Miao et al., 2023) and new verification algorithms for the multi-draft scenario are proposed (Li et al., 2024; Chen et al., 2024). While increasing the number of draft sequences has shown to improve the overall speedup, it comes at the cost of more computation. In Appendix D.1, we show that in the large batch setting, where the inference is less memory bound, our method outperforms these methods without increasing the number of draft blocks. In all of these works, the verification algorithm is a generalization of the token verification procedure. Extending block verification to the multi-sample case is an interesting future direction.

## 8 DISCUSSION

We showed that the standard *token verification* algorithm used by speculative decoding is not optimal. Further, we proposed a new verification algorithm, *block verification* and proved that it is an optimal verification algorithm. We also demonstrated empirically that block verification consistently outperforms token verification in a range of tasks. While the theoretical proofs are somewhat involved, the actual implementation of block verification is not more complex than the standard algorithm (see Appendix A), and since our proposed algorithm can only perform better, never worse, than the standard token verification algorithm (see Theorem 2), it can be used as a good default in speculative decoding implementations.

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

# A  PYTHON IMPLEMENTATION

In this section we provide a sketch implementation of block verification (Algorithm 2) in Python. Note that these are meant for illustration purposes only and are not fit for practical use.

Let $V = |\mathcal{X}|$ be the size of the vocabulary. The inputs to the algorithm are:

- `ps`: an $(\gamma+1) \times V$ numpy array with the distributions from the large model $\mathcal{M}_b(\cdot \mid c, X^i)$;
- `qs`: an $\gamma \times V$ numpy array with the distributions from the draft model $\mathcal{M}_s(\cdot \mid c, X^i)$;
- `drafts`: a length-$\gamma$ numpy array with the ids of the draft tokens $X^\gamma$;

```python
def block_verification(
  ps: np.ndarray, qs: np.ndarray, drafts: np.ndarray) -> list[int]:
  draft_length, vocab_size = qs.shape
  qs.resize((draft_length+1, vocab_size)) # Append a zero vector
  token_sequence = [] # Will include the token sequence we return
  accept_probability = 1.0 # Acceptance prob. for each sub-block
  probability_ratios = ps / qs
  # Add one token to indicate rejecting the sequence
  vocab_plus_one = np.arange(vocab_size + 1)
  for token_index, token_value in enumerate(xs):
    # Unnormalized residual probability
    sampling_weights[:vocab_size] = np.maximum(
        0, ps[token_index] * accept_probability - qs[token_index])
    # Unnormalized probability of rejecting the sequence
    sampling_weights[vocab_size] = 1 - accept_probability
    sampling_weights /= np.sum(sampling_weights)
    chosen_token = np.random.choice(vocab_plus_one,
                                    p=sampling_weights)
    # Update the sequence
    if chosen_token < vocab_size:
      token_sequence = xs[:token_index] + [chosen_token]
    # Update the acceptance probability
    accept_probability = min(1, probability_ratios[
        token_index, token_value] * accept_probability)
  return token_sequence
```

For reference, here is a sketch implementation of the token verification algorithm (Algorithm 1):

```python
def token_verification(
  ps: np.ndarray, qs: np.ndarray, drafts: np.ndarray) -> list[int]:
  draft_length, vocab_size = qs.shape
  qs.resize((draft_length+1, vocab_size)) # Append a zero vector.
  token_sequence = [] # Will include the token sequence we return
  probability_ratios = ps / qs
  token_index = 0
  vocab_range = np.arange(vocab_size)
  for token_value in xs:
    accept_probability = probability_ratios[token_index, token_value]
    if (not np.isfinite(accept_probability) or
        np.random.random() > accept_probability): # Rejection
      break
    token_index += 1
    token_sequence.append(token_value)
  # Calculate the residual distribution
  sampling_weights = np.maximum(0, ps[token_index] - qs[token_index])
  sampling_weights /= np.sum(sampling_weights)
```

```
token_sequence.append(np.random.choice(vocab_range,
                                        p=sampling_weights))
    return token_sequence
```

## B    FORMAL PROOFS

We start by setting up a few necessary notations. Let $\mathcal{X}$ be the space of output tokens. For $\ell > 1$, we use $\mathcal{M}^\ell(\cdot \mid c)$ to denote the joint distribution of the next $\ell$ tokens conditioned on the prefix under $\mathcal{M}$, *i.e.*, for all $x_1, \ldots x_\ell \in \mathcal{X}^\ell$, $\mathcal{M}^\ell(x_1, \ldots, x_\ell \mid c) = \prod_{i=1}^\ell \mathcal{M}(x_i \mid c, x^{i-1})$. We use $\mathcal{M}^*(\cdot \mid c)$ to denote the distribution of the sequence up to the end of the generative process. Below we first describe a necessary and sufficient condition for a valid draft verification algorithm in Algorithm 3.

**Lemma 2.** $\forall c, \mathcal{M}_s, \mathcal{M}_b, \gamma$, let $X^\gamma$ be generated from $\mathcal{M}_s^\gamma(\cdot \mid c)$, and

$$X^\tau, Y = \text{VERIFY}(X^\gamma, \{\mathcal{M}_s(\cdot \mid c, X^i)\}_{i=0}^{\gamma-1}, \{\mathcal{M}_b(\cdot \mid c, X^i)\}_{i=0}^\gamma).$$

*Let $Z^{\gamma-\tau}$ be generated from $\mathcal{M}_b^{\gamma-\tau}(\cdot \mid c, X^\tau, Y)$.*

VERIFY *is a valid draft verification algorithm (Definition 1) if and only if $\forall c, \mathcal{M}_s, \mathcal{M}_b, \gamma$,*

$$X^\tau, Y, Z^{\gamma-\tau} \sim_{\text{p}} \mathcal{M}_b^{\gamma+1}(\cdot \mid c). \tag{7}$$

*Proof.* We first prove the forward direction (Equation (7) implies that VERIFY satisfies Definition 1) by induction on the maximum generation length of $\mathcal{M}_b(\cdot \mid c)$. When the maximum generation length is 0, for all new context $c'$, we have the next token is a point mass over E.O.S, *i.e.*,

$$\mathcal{M}_b(x \mid c, c') = \delta\{x = \text{E.O.S}\}.$$

Then Equation (7) implies that VERIFY will only output E.O.S, which is the same as Definition 1. Suppose Equation (6) holds for all context and $\mathcal{M}_b$ with generation length at most $T$, for a context $c$ and $\mathcal{M}_b$ with maximum generation length at most $T + 1$, we have that the output of SPECDEC$(c, \mathcal{M}_b, \mathcal{M}_s, \gamma, \text{VERIFY})$ is

$$X^\tau, Y, \text{SPECDEC}((c, X^\tau, Y), \mathcal{M}_b, \mathcal{M}_s, \gamma, \text{VERIFY}).$$

Let $Z^{\gamma-\tau}$ be the first $\gamma - \tau$ tokens from SPECDEC$((c, X^\tau, Y), \mathcal{M}_b, \mathcal{M}_s, \gamma, \text{VERIFY})$, and $O^*$ be the tokens after. Since $X^\tau, Y$ is at least of length one, the generation length of $\mathcal{M}_b(\cdot \mid c, X^\tau, Y)$ is at most $T$. By the induction hypothesis, we have

$$Z^{\gamma-\tau} \sim_{\text{p}} \mathcal{M}_b^{\gamma-\tau}(\cdot \mid c, X^\tau, Y),$$

and

$$O^* \sim_{\text{p}} \mathcal{M}_b^*(\cdot \mid c, X^\tau, Y, Z^{\gamma-\tau}).$$

And hence by Equation (7),

$$\begin{aligned}
\text{SPECDEC}(c, \mathcal{M}_b, \mathcal{M}_s, \gamma, \text{VERIFY}) &= X^\tau, Y, \text{SPECDEC}((c, X^\tau, Y), \mathcal{M}_b, \mathcal{M}_s, \gamma, \text{VERIFY}) \\
&= X^\tau, Y, Z^{\gamma-\tau}, O^* \\
&\sim_{\text{p}} \mathcal{M}_b^*(\cdot \mid c).
\end{aligned}$$

This completes the proof for the forward direction.

For the backward direction, we have Equation (6) implies that for all $X^\tau, Y$,

$$\text{SPECDEC}((c, X^\tau, Y), \mathcal{M}_b, \mathcal{M}_s, \gamma, \text{VERIFY})[: \gamma - \tau]^5 \sim_{\text{p}} \mathcal{M}_b^{\gamma-\tau}(\cdot \mid c, X^\tau, Y).$$

Let $Z^{\gamma-\tau}$ be a draw from $\mathcal{M}_b^{\gamma-\tau}(\cdot \mid c, X^\tau, Y)$, then

$$Z^{\gamma-\tau} \sim_{\text{p}} \text{SPECDEC}((c, X^\tau, Y), \mathcal{M}_b, \mathcal{M}_s, \gamma, \text{VERIFY})[: \gamma - \tau].$$

---

[5]We use $v[i : j]$ to denote the entries $i$ to $j$ in $v$.

And hence when $X^\tau, Y$ is the output of VERIFY,

$$
\begin{aligned}
X^\tau, Y, Z^{\gamma-\tau} &\sim_{\mathrm{p}} X^\tau, Y, \text{SPECDEC}((\boldsymbol{c}, X^\tau, Y), \mathcal{M}_b, \mathcal{M}_s, \gamma, \text{VERIFY})[:\gamma - \tau] \\
&\sim_{\mathrm{p}} \text{SPECDEC}(\boldsymbol{c}, \mathcal{M}_b, \mathcal{M}_s, \gamma, \text{VERIFY})[:\gamma + 1] \\
&\sim_{\mathrm{p}} \mathcal{M}_b^{\gamma+1}(\cdot \mid \boldsymbol{c}),
\end{aligned}
$$

where the last derivation follows from Equation (6) in Definition 1. $\qquad\square$

In all proofs below, we fix the context $\boldsymbol{c}$, and the models $\mathcal{M}_s$ and $\mathcal{M}_b$. We note that the proofs won't use specific information about these choices and hence can be easily extended to all cases.

## B.1 PROOF OF THEOREM 1

By Lemma 2, it would be enough to prove that block verification satisfies Equation (7). For simplicity, we often refer to the sequence $(X^\tau, Y, Z^{\gamma-\tau})$ by $O^{\gamma+1}$. Note that $O_{\gamma+1} \sim \mathcal{M}_b(\cdot \mid \boldsymbol{c}, O^\gamma)$ always holds since when $\tau < \gamma$, $O_{\gamma+1} \sim \mathcal{M}_b(\cdot \mid \boldsymbol{c}, O^\gamma)$ by definition and when $\tau = \gamma$, $O_{\gamma+1} = Y \sim \mathcal{M}_b(\cdot \mid \boldsymbol{c}, X^\gamma) = \mathcal{M}_b(\cdot \mid \boldsymbol{c}, O^\gamma)$. Hence it is enough to prove the following

$$
\forall \ell \leq \gamma, \forall x^\ell \in \mathcal{X}^\ell, \quad \Pr\left(O^\ell = x^\ell\right) = \mathcal{M}_b^\ell(x^\ell \mid \boldsymbol{c}), \tag{8}
$$

Note that in block verification, $p_i$'s depend on the draft tokens $X^\gamma$. The following definition makes this explicit. Let $p_i$ be such that $p_0 = 1$, and $\forall 1 \leq i \leq \gamma, x^i \in \mathcal{X}^i$,

$$
p_i(x^i \mid \boldsymbol{c}) = \min\left\{ p_{i-1}(x^i \mid \boldsymbol{c}) \frac{\mathcal{M}_b(x_i \mid \boldsymbol{c}, x^{i-1})}{\mathcal{M}_s(x_i \mid \boldsymbol{c}, x^{i-1})}, 1 \right\}. \tag{9}
$$

For most cases, when the prefix $\boldsymbol{c}$ is clear, we will ignore $\boldsymbol{c}$ and simply use $p_i(x^i) = p_i(x^i \mid \boldsymbol{c})$. We will only make the prefix explicit when necessary.

We first state the following lemma on the distribution of the number of tokens accepted by block verification.

**Lemma 3.** *Let $X^\gamma \sim \mathcal{M}_s^\gamma(\cdot \mid \boldsymbol{c})$, and*

$$
X^\tau, Y = \text{BLOCKVERIFY}(X^\gamma, \{\mathcal{M}_s(\cdot \mid \boldsymbol{c}, X^i)\}_{i=0}^{\gamma-1}, \{\mathcal{M}_b(\cdot \mid \boldsymbol{c}, X^i)\}_{i=0}^\gamma).
$$

*Then we have $\forall i \leq \gamma$, and $x^i \in \mathcal{X}^i$,*

$$
\Pr\left(\tau \geq i \mid X^i = x^i\right) = p_i(x^i).
$$

We first prove Theorem 1 based on Lemma 3 and defer the proof of the lemma to Appendix B.3. We prove Equation (8) by induction on the time index $\ell$. When $\ell = 1$, $O_1$ is either $X_1$, or a residual sample from $p_{\text{res}}^{\text{block}}(\cdot \mid \boldsymbol{c})$ where

$$
p_{\text{res}}^{\text{block}}(\cdot \mid \boldsymbol{c}) = \frac{\max\{\mathcal{M}_b(x \mid \boldsymbol{c}) - \mathcal{M}_s(x \mid \boldsymbol{c}), 0\}}{\sum_{x'} \max\{\mathcal{M}_b(x' \mid \boldsymbol{c}) - \mathcal{M}_s(x' \mid \boldsymbol{c}), 0\}},
$$

Hence we have $\forall x \in \mathcal{X}$, by Lemma 3,

$$
\begin{aligned}
&\Pr\left(O_1 = x\right) \\
&= \Pr\left(O_1 = x, \tau \geq 1\right) + \Pr\left(O_1 = x, \tau = 0\right) \\
&= \Pr\left(X_1 = x\right)\Pr\left(\tau \geq 1 \mid X_1 = x\right) + \sum_{x'} \Pr\left(X_1 = x'\right)(1 - \Pr\left(\tau \geq 1 \mid X_1 = x'\right)) \cdot p_{\text{res}}^{\text{block}}(x \mid \boldsymbol{c}) \\
&= \mathcal{M}_s(x \mid \boldsymbol{c}) \cdot p_1(x) + \sum_{x'} \mathcal{M}_s(x' \mid \boldsymbol{c})(1 - p_1(x')) \cdot p_{\text{res}}^{\text{block}}(x \mid \boldsymbol{c}) \\
&= \min\{\mathcal{M}_b(x \mid \boldsymbol{c}), \mathcal{M}_s(x \mid \boldsymbol{c})\} + \sum_{x'} \max\{\mathcal{M}_b(x' \mid \boldsymbol{c}) - \mathcal{M}_s(x' \mid \boldsymbol{c}), 0\} \cdot p_{\text{res}}^{\text{block}}(x \mid \boldsymbol{c}) \qquad (10) \\
&= \min\{\mathcal{M}_b(x \mid \boldsymbol{c}), \mathcal{M}_s(x \mid \boldsymbol{c})\} + \max\{\mathcal{M}_b(x \mid \boldsymbol{c}) - \mathcal{M}_s(x \mid \boldsymbol{c}), 0\} \qquad (11) \\
&= \mathcal{M}_b(x \mid \boldsymbol{c}),
\end{aligned}
$$

where Equation (10) comes from the definition of $p_1$ in Equation (9) and Equation (11) is due to Equation (4) with $i = 0$. Hence the Equation (8) holds for $\ell = 1$. Suppose Equation (8) holds up to $\ell < \gamma$. For $\ell = \ell + 1$, we have $O_{\ell+1}$ is either equal to $X_{\ell+1}$ when $\tau \geq \ell + 1$, or a sample from $p_{\text{res}}^{\text{block}}(\cdot \mid \boldsymbol{c}, X^{\ell})$ when $\tau = \ell$, or a sample from $\mathcal{M}_b(\cdot \mid \boldsymbol{c}, O^{\ell})$ when $\tau < \ell$. Hence $\Pr\left(O^{\ell+1} = x^{\ell+1}\right)$ can be broken down below:

$$\Pr\left(O^{\ell+1} = x^{\ell+1}\right) =$$
$$\Pr\left(O^{\ell+1} = x^{\ell+1}, \tau \geq \ell + 1\right) + \Pr\left(O^{\ell+1} = x^{\ell+1}, \tau = \ell\right) + \Pr\left(O^{\ell+1} = x^{\ell+1}, \tau < \ell\right) \quad (12)$$

For the first term ($\tau \geq \ell + 1$), we have

$$\Pr\left(O^{\ell+1} = x^{\ell+1}, \tau \geq \ell + 1\right)$$
$$= \Pr\left(X^{\ell+1} = x^{\ell+1}\right) \cdot \Pr\left(\tau \geq \ell + 1 \mid X^{\ell+1} = x^{\ell+1}\right)$$
$$= \mathcal{M}_s(x^{\ell+1} \mid \boldsymbol{c}) \cdot p_{\ell+1}(x^{\ell+1})$$
$$= \mathcal{M}_s(x^{\ell} \mid \boldsymbol{c}) \cdot \min\{p_\ell(x^\ell)\mathcal{M}_b(x_{\ell+1} \mid \boldsymbol{c}, x^\ell), \mathcal{M}_s(x_{\ell+1} \mid \boldsymbol{c}, x^\ell)\}. \quad (13)$$

For the second term ($\tau = \ell$), we have

$$\Pr\left(O^{\ell+1} = x^{\ell+1}, \tau = \ell\right)$$
$$= \Pr\left(X^\ell = x^\ell\right) \cdot \Pr\left(\tau = \ell \mid X^\ell = x^\ell\right) \cdot \Pr\left(O_{\ell+1} = x_{\ell+1} \mid O^\ell = x^\ell, \tau = \ell\right)$$
$$= \Pr\left(X^\ell = x^\ell\right) \cdot \Pr\left(\tau = \ell \mid X^\ell = x^\ell\right) \cdot p_{\text{res}}^{\text{block}}(x_{\ell+1} \mid \boldsymbol{c}, x^\ell)$$

Note that,

$$\Pr\left(\tau = \ell \mid X^\ell = x^\ell\right)$$
$$= \Pr\left(\tau \geq \ell \mid X^\ell = x^\ell\right) - \sum_x \mathcal{M}_s(x \mid \boldsymbol{c}, x^\ell) \cdot \Pr\left(\tau \geq \ell + 1 \mid \boldsymbol{c}, X^{\ell+1} = x^\ell, x\right)$$
$$= p_\ell(x^\ell) - \sum_x \mathcal{M}_s(x \mid \boldsymbol{c}, x^\ell) \cdot p_{\ell+1}(x^\ell, x)$$
$$= p_\ell(x^\ell) - \sum_x \min\{p_\ell(x^\ell)\mathcal{M}_b(x \mid \boldsymbol{c}, x^\ell), \mathcal{M}_s(x \mid \boldsymbol{c}, x^\ell)\}$$
$$= \sum_x \max\{p_\ell(x^\ell)\mathcal{M}_b(x \mid \boldsymbol{c}, x^\ell) - \mathcal{M}_s(x \mid \boldsymbol{c}, x^\ell), 0\}.$$

And hence by the definition of $p_{\text{res}}^{\text{block}}(x_{\ell+1} \mid \boldsymbol{c}, x^\ell)$ in Equation (4), we have

$$\Pr\left(O^{\ell+1} = x^{\ell+1}, \tau = \ell\right)$$
$$= \mathcal{M}_s(x^\ell \mid \boldsymbol{c}) \cdot \max\{p_\ell(x^\ell)\mathcal{M}_b(x_{\ell+1} \mid \boldsymbol{c}, x^\ell) - \mathcal{M}_s(x_{\ell+1} \mid \boldsymbol{c}, x^\ell), 0\}. \quad (14)$$

For the third term ($\tau < \ell$), by induction, and the generation process of $O^{\gamma+1}$, we have

$$\Pr\left(O^{\ell+1} = x^{\ell+1}, \tau < \ell\right) = \Pr\left(O^\ell = x^\ell, \tau < \ell\right) \cdot \Pr\left(O_{\ell+1} = x_{\ell+1} \mid O^\ell = x^\ell, \tau < \ell\right)$$
$$= \left(\Pr\left(O^\ell = x^\ell\right) - \Pr\left(O^\ell = x^\ell, \tau \geq \ell\right)\right) \cdot \mathcal{M}_b(x_{\ell+1} \mid \boldsymbol{c}, x^\ell)$$
$$= \left(\mathcal{M}_b(x^\ell \mid \boldsymbol{c}) - \mathcal{M}_s(x^\ell \mid \boldsymbol{c})p_\ell(x^\ell)\right) \cdot \mathcal{M}_b(x_{\ell+1} \mid \boldsymbol{c}, x^\ell) \quad (15)$$

Plugging Equations (13) to (15) into Equation (12), we get $\forall x^{\ell+1} \in \mathcal{X}^{\ell+1}$,

$$\Pr\left(O^{\ell+1} = x^{\ell+1}\right) = \mathcal{M}_b(x^{\ell+1} \mid \boldsymbol{c}),$$

completing the induction step and hence the proof of Equation (8) and Theorem 1.

## B.2 PROOF OF THEOREM 2

We first state the following lemma, which when combined with Lemma 3, shows that in one iteration, among all valid draft verification algorithms, block verification accepts each subsequence with the highest probability.

**Lemma 4.** *For draft verification algorithms that satisfy the constraints in Lemma 2, we have $\forall i \leq \gamma$, and $x^i \in \mathcal{X}^i$,*

$$\Pr\left(\tau \geq i \mid X^i = x^i\right) \leq p_i(x^i).$$

We defer the proof of the lemma to Appendix B.4 and first prove Theorem 2 based on the lemma.

We start by breaking down the expected number of decoded tokens $\mathbb{E}_{\text{VERIFY}}[N(i)]$ into the distribution of $N(i)$ on different sample paths. Let $O^* = O_1, O_2, \ldots,$ be the complete output sequence from speculative decoding. We set all tokens after E.O.S to be E.O.S as well. Then we have

$$\mathbb{E}_{\text{VERIFY}}[N(i)] = \sum_{\ell=1}^{\infty} \Pr_{\text{VERIFY}}\left(N(i) \geq \ell\right) = \sum_{x^* \in \mathcal{X}^*} \sum_{\ell=1}^{\infty} \Pr_{\text{VERIFY}}\left(O^* = x^*, N(i) \geq \ell\right)$$

Hence it would be enough to prove the following.

**Lemma 5.** *For all draft verification algorithms that satisfy the constraints in Lemma 2, we have $\forall c$, and output $x^* \in \mathcal{X}^*$*

$$\Pr_{\text{VERIFY}}\left(O = x^*, N(i) \geq \ell \mid c\right) \leq \Pr_{\text{BLOCKVERIFY}}\left(O = x^*, N(i) \geq \ell \mid c\right) \tag{16}$$

We prove the lemma by induction on the number of iterations $i$. We first prove the following lemma for all verification algorithms.

**Lemma 6.** *For all $\ell \leq \gamma$,*

$$\Pr_{\text{VERIFY}}\left(O^* = x^*, \tau \geq \ell \mid c\right) = \Pr_{\text{VERIFY}}\left(O^\ell = x^\ell, \tau \geq \ell \mid c\right) \cdot \mathcal{M}_b^*\left(x^{\ell+1:*} \mid c, x^\ell\right).$$

*Proof.* When $\Pr_{\text{VERIFY}}\left(O^\ell = x^\ell, \tau \geq \ell \mid c\right) = 0$, the bound is trivial since both sides are 0. Otherwise we have

$$\Pr_{\text{VERIFY}}\left(O^* = x^*, \tau \geq \ell \mid c\right)$$
$$= \Pr_{\text{VERIFY}}\left(O^{\ell+1:*} = x^{\ell+1:*}, O^\ell = x^\ell, \tau \geq \ell \mid c\right)$$
$$= \Pr_{\text{VERIFY}}\left(O^\ell = x^\ell, \tau \geq \ell \mid c\right) \cdot \Pr_{\text{VERIFY}}\left(O^{\ell+1:*} = x^{\ell+1:*} \mid O^\ell = x^\ell, \tau \geq \ell, c\right)$$

It would be enough to show that

$$\Pr_{\text{VERIFY}}\left(O^{\ell+1:*} = x^{\ell+1:*} \mid O^\ell = x^\ell, \tau \geq \ell, c\right) = \mathcal{M}_b^*\left(x^{\ell+1:*} \mid c, x^\ell\right). \tag{17}$$

Note that

$$\mathcal{M}_b^*\left(x^{\ell+1:*} \mid c, x^\ell\right)$$
$$= \Pr_{\text{VERIFY}}\left(O^{\ell+1:*} = x^{\ell+1:*} \mid O^\ell = x^\ell, c\right)$$
$$= \Pr_{\text{VERIFY}}\left(O^{\ell+1:*} = x^{\ell+1:*}, \tau \geq \ell \mid O^\ell = x^\ell, c\right) + \Pr_{\text{VERIFY}}\left(O^{\ell+1:*} = x^{\ell+1:*}, \tau < \ell \mid O^\ell = x^\ell, c\right)$$
$$= \Pr_{\text{VERIFY}}\left(\tau \geq \ell \mid O^\ell = x^\ell, c\right) \Pr_{\text{VERIFY}}\left(O^{\ell+1:*} = x^{\ell+1:*} \mid O^\ell = x^\ell, \tau \geq \ell, c\right)$$
$$\quad + \Pr_{\text{VERIFY}}\left(\tau < \ell \mid O^\ell = x^\ell, c\right) \Pr_{\text{VERIFY}}\left(O^{\ell+1:*} = x^{\ell+1:*} \mid O^\ell = x^\ell, \tau < \ell, c\right) \tag{18}$$

When $\Pr_{\text{VERIFY}}\left(\tau < \ell \mid O^\ell = x^\ell, c\right) = 0$, we have $\Pr_{\text{VERIFY}}\left(\tau \geq \ell \mid O^\ell = x^\ell, c\right) = 1$, and hence

$$\Pr_{\text{VERIFY}}\left(O^{\ell+1:*} = x^{\ell+1:*} \mid O^\ell = x^\ell, \tau \geq \ell, c\right) = \mathcal{M}_b^*\left(x^{\ell+1:*} \mid c, x^\ell\right).$$

When $\Pr_{\text{VERIFY}}\left(\tau < \ell \mid O^\ell = x^\ell, c\right) > 0$, since VERIFY is a valid verification algorithm (Definition 1), we have tokens starting from location $\ell + 1$ is valid draw from $\mathcal{M}_b(\cdot \mid c, x^\ell)$, *i.e.,*

$$\Pr_{\text{VERIFY}}\left(O^{\ell+1:*} = x^{\ell+1:*} \mid O^\ell = x^\ell, \tau < \ell, c\right) = \mathcal{M}_b^*\left(x^{\ell+1:*} \mid c, x^\ell\right).$$

Plugging this into Equation (18) completes the proof. □

When $i = 1$, we have that $N(1) = \tau + 1$, where $\tau$ is the number of accepted tokens. Hence we have

$$\Pr_{\text{VERIFY}} \left( O^* = x^*, N(1) \geq \ell \mid \boldsymbol{c} \right)$$

$$= \Pr_{\text{VERIFY}} \left( O^* = x^*, \tau \geq \ell - 1 \mid \boldsymbol{c} \right)$$

$$= \Pr_{\text{VERIFY}} \left( O^{\ell-1} = x^{\ell-1}, \tau \geq \ell - 1 \mid \boldsymbol{c} \right) \cdot \mathcal{M}_b^* \left( x^{\ell:*} \mid \boldsymbol{c}, x^{\ell-1} \right) \qquad \triangleleft \text{Lemma } 6$$

$$= \mathcal{M}_s(x^{\ell-1} \mid \boldsymbol{c}) \mathcal{M}_b^*(x^{\ell:*} \mid \boldsymbol{c}, x^{\ell-1}) \Pr_{\text{VERIFY}} \left( \tau \geq \ell - 1 \mid X^{\ell-1} = x^{\ell-1}, \boldsymbol{c} \right),$$

where the last equality is because $\Pr_{\text{VERIFY}} \left( O^{\ell-1} = x^{\ell-1}, \tau \geq \ell - 1 \mid \boldsymbol{c} \right)$ is the probability of the event that $O^{\ell-1} = x^{\ell-1}$ is contained in the accepted tokens, which happens under the joint of two events: (1) The first $\ell - 1$ tokens in the draft block from the small model $X^{\ell-1} = x^{\ell-1}$. This probability is $\mathcal{M}_s(x^{\ell-1} \mid \boldsymbol{c})$; (2) Conditioned on $X^{\ell-1} = x^{\ell-1}$, at least $\ell - 1$ tokens are accepted, this is $\Pr_{\text{VERIFY}} \left( \tau \geq \ell - 1 \mid X^{\ell-1} = x^{\ell-1}, \boldsymbol{c} \right)$, and hence

$$\Pr_{\text{VERIFY}} \left( O^{\ell-1} = x^{\ell-1}, \tau \geq \ell - 1 \mid \boldsymbol{c} \right) = \mathcal{M}_s(x^{\ell-1} \mid \boldsymbol{c}) \cdot \Pr_{\text{VERIFY}} \left( \tau \geq \ell - 1 \mid X^{\ell-1} = x^{\ell-1}, \boldsymbol{c} \right).$$

Similarly, we have

$$\Pr_{\text{BLOCKVERIFY}} \left( O^* = x^*, N(1) \geq \ell \mid \boldsymbol{c} \right)$$

$$= \mathcal{M}_s(x^{\ell-1} \mid \boldsymbol{c}) \mathcal{M}_b^*(x^{\ell:*} \mid \boldsymbol{c}, x^{\ell-1}) \Pr_{\text{BLOCKVERIFY}} \left( \tau \geq \ell - 1 \mid X^{\ell-1} = x^{\ell-1}, \boldsymbol{c} \right).$$

Note that Lemmas 3 and 4 imply that for all verification algorithm, we have

$$\Pr_{\text{VERIFY}} \left( \tau \geq \ell - 1 \mid X^{\ell-1} = x^{\ell-1}, \boldsymbol{c} \right) \leq \Pr_{\text{BLOCKVERIFY}} \left( \tau \geq \ell - 1 \mid X^{\ell-1} = x^{\ell-1}, \boldsymbol{c} \right).$$

Combining these, we have

$$\Pr_{\text{VERIFY}} \left( O^* = x^*, N(1) \geq \ell \mid \boldsymbol{c} \right) \leq \Pr_{\text{BLOCKVERIFY}} \left( O^* = x^*, N(1) \geq \ell \mid \boldsymbol{c} \right). \tag{19}$$

Suppose the lemma holds for all iterations up to $i$, for the $(i+1)$th iteration, let $\tau_{i+1}$ be the number of tokens accepted in the $(i+1)$th iteration, we have

$$\Pr_{\text{VERIFY}} \left( O^* = x^*, N(i+1) \geq \ell \mid \boldsymbol{c} \right)$$

$$= \sum_{\ell' < \ell} \Pr_{\text{VERIFY}} \left( O^* = x^*, N(i) = \ell', N(i+1) \geq \ell \mid \boldsymbol{c} \right)$$

$$= \sum_{\ell' < \ell} \Pr_{\text{VERIFY}} \left( O^* = x^*, N(i) = \ell' \mid \boldsymbol{c} \right) \Pr_{\text{VERIFY}} \left( \tau_{i+1} \geq \ell - \ell' - 1 \mid \boldsymbol{c}, O^* = x^*, N(i) = \ell' \right)$$

$$= \Pr_{\text{VERIFY}} \left( O^* = x^* \mid \boldsymbol{c} \right) \sum_{\ell' < \ell} \Pr_{\text{VERIFY}} \left( N(i) = \ell' \mid O^* = x^*, \boldsymbol{c} \right)$$

$$\cdot \Pr_{\text{VERIFY}} \left( \tau_{i+1} \geq \ell - \ell' - 1 \mid \boldsymbol{c}, O^* = x^*, N(i) = \ell' \right)$$

$$= \mathcal{M}_b(x^* \mid \boldsymbol{c}) \sum_{\ell' < \ell} \Pr_{\text{VERIFY}} \left( N(i) = \ell' \mid O^* = x^*, \boldsymbol{c} \right) \Pr_{\text{VERIFY}} \left( \tau_{i+1} \geq \ell - \ell' - 1 \mid \boldsymbol{c}, O^* = x^*, N(i) = \ell' \right)$$

$$\tag{20}$$

Let $\eta_{\text{VERIFY}}$ be a random variable distributed according to $\Pr_{\text{VERIFY}} \left( N(i) = \ell' \mid O^* = x^*, \boldsymbol{c} \right)$, and

$$f_{\text{VERIFY}}(\eta) = \Pr_{\text{VERIFY}} \left( \tau_{i+1} \geq \ell - \eta - 1 \mid \boldsymbol{c}, O^* = x^*, N(i) = \eta \right).$$

Plugging these into Equation (20), we have

$$\Pr_{\text{VERIFY}} \left( O^* = x^*, N(i+1) \geq \ell \mid \boldsymbol{c} \right) = \mathcal{M}_b(x^* \mid \boldsymbol{c}) \mathbb{E}_{\eta_{\text{VERIFY}}} \left[ f_{\text{VERIFY}}(\eta_{\text{VERIFY}}) \right]$$

Note that let $\boldsymbol{c}_\eta = \boldsymbol{c}, x^\eta$, we have

$$f_{\text{VERIFY}}(\eta) = \Pr_{\text{VERIFY}} \left( \tau_{i+1} \geq \ell - \eta - 1 \mid \boldsymbol{c}, O^* = x^*, N(i) = \eta \right)$$

$$= \Pr_{\text{VERIFY}} \left( \tau \geq \ell - \eta - 1 \mid \boldsymbol{c}_\eta, O^* = x^{\eta+1:*} \right) \tag{21}$$

where Equation (21) is due to the iterative structure of speculative decoding and after generating $O^\eta = x^\eta$ in the first $i$ iterations ($N(i) = \eta$), the next iteration is the same as generating from scratch with context $\boldsymbol{c}_\eta = \boldsymbol{c}, x^\eta$. Similarly, we have

$$f_{\text{BLOCKVERIFY}}(\eta) = \Pr_{\text{BLOCKVERIFY}} \left( \tau \geq \ell - \eta - 1 \mid \boldsymbol{c}_\eta, O^* = x^{\eta+1:*} \right).$$

Note that $\forall \boldsymbol{c}, x^* \in \mathcal{X}^*$, and $i$, we have

$$
\begin{aligned}
\Pr_{\text{BLOCKVERIFY}} \left( \tau \geq i \mid \boldsymbol{c}, O^* = x^* \right) &= \frac{\Pr_{\text{BLOCKVERIFY}} \left( O^* = x^*, \tau \geq i \mid \boldsymbol{c} \right)}{\Pr_{\text{BLOCKVERIFY}} \left( O^* = x^* \mid \boldsymbol{c} \right)} \\
&= \frac{\Pr_{\text{BLOCKVERIFY}} \left( O^* = x^*, \tau \geq i \mid \boldsymbol{c} \right)}{\mathcal{M}_b^*(x^* \mid \boldsymbol{c})} \\
&\geq \frac{\Pr_{\text{VERIFY}} \left( O^* = x^*, \tau \geq i \mid \boldsymbol{c} \right)}{\mathcal{M}_b^*(x^* \mid \boldsymbol{c})} \quad (22) \\
&= \Pr_{\text{VERIFY}} \left( \tau \geq i \mid \boldsymbol{c}, O^* = x^* \right), \quad (23)
\end{aligned}
$$

where Equation (22) is due to Equation (19) and that $N(1) = \tau + 1$. Equation (23) implies that $f_{\text{BLOCKVERIFY}}(\eta) \geq f_{\text{VERIFY}}(\eta)$, and hence we have

$$
\begin{aligned}
\Pr_{\text{VERIFY}} \left( O^* = x^*, N(i+1) \geq \ell \mid \boldsymbol{c} \right) &= \mathcal{M}_b(x^* \mid \boldsymbol{c}) \mathbb{E}_{\eta_{\text{VERIFY}}} \left[ f_{\text{VERIFY}}(\eta) \right] \\
&\leq \mathcal{M}_b(x^* \mid \boldsymbol{c}) \mathbb{E}_{\eta_{\text{VERIFY}}} \left[ f_{\text{BLOCKVERIFY}}(\eta) \right].
\end{aligned}
$$

Note that $\Pr_{\text{BLOCKVERIFY}} \left( O^* = x^*, N(i+1) \geq \ell \mid \boldsymbol{c} \right) = \mathcal{M}_b(x^* \mid \boldsymbol{c}) \mathbb{E}_{\eta_{\text{BLOCKVERIFY}}} \left[ f_{\text{BLOCKVERIFY}}(\eta) \right]$. It would be enough to prove that

$$\mathbb{E}_{\eta_{\text{VERIFY}}} \left[ f_{\text{BLOCKVERIFY}}(\eta) \right] \leq \mathbb{E}_{\eta_{\text{BLOCKVERIFY}}} \left[ f_{\text{BLOCKVERIFY}}(\eta) \right]. \quad (24)$$

Next we prove Equation (24) using the lemma below.

**Lemma 7** (Quirk and Saposnik (1962)). *Let $f : \mathbb{R} \to \mathbb{R}$ be an increasing function and $X_1$ stochastically dominates $X_2$, meaning $\forall x$, we have $\Pr(X_1 \geq x) \geq \Pr(X_2 \geq x)$, then we have*

$$\mathbb{E}[f(X_1)] \geq \mathbb{E}[f(X_2)].$$

By the induction hypothesis, we have $\eta_{\text{BLOCKVERIFY}}$ stochastically dominates (Quirk and Saposnik, 1962) $\eta_{\text{VERIFY}}$ for any valid verification algorithm. It remains to show that $f_{\text{BLOCKVERIFY}}(\eta)$ is an increasing function. By definition, since $0 \leq \tau \leq \gamma$, when $\eta < \ell - \gamma - 1$, $f_{\text{BLOCKVERIFY}}(\eta) = 0$ and when $\eta > \ell - 1$, $f_{\text{BLOCKVERIFY}}(\eta) = 1$. When $\ell - \gamma - 1 \leq \eta \leq \ell - 2$, by definition and Lemma 3, $f_{\text{BLOCKVERIFY}}(\eta) = p_{\ell-\eta-1}(x^{\eta+1:\ell-1} \mid \boldsymbol{c}, x^\eta)$. To see that $p_{\ell-\eta-1}(x^{\eta+1:\ell-1} \mid \boldsymbol{c}, x^\eta)$ is an increasing function of $\eta$, for $\eta' = \eta + 1$, we can obtain $p_{\ell-\eta'-1}(x^{\eta'+1:\ell-1} \mid \boldsymbol{c}, x^{\eta'})$ by following the same recursion steps as in Equation (9) but replacing $p_1(x^{\eta+1:\ell-1} \mid \boldsymbol{c}, x^\eta)$ with $p_0(x^{\eta+2:\ell-1} \mid \boldsymbol{c}, x^{\eta+1}) = 1$, and hence only increasing the values.

This proves that $f_{\text{BLOCKVERIFY}}$ is increasing and hence Equation (24) holds. This implies that the induction step holds due to Lemma 7, completing proof of Theorem 2.

### B.3 PROOF OF LEMMA 3

Note that in Line 4 of Algorithm 2, $p_i = p_i(X^i)$. We prove the statement by backward induction. When $i = \gamma$, we have by definition of $h_\gamma^{\text{block}}$ in Figure 2, $\forall x^\gamma \in \mathcal{X}^\gamma$,

$$\Pr \left( \tau \geq \gamma \mid X^\gamma = x^\gamma \right) = h_\gamma^{\text{block}} = p_\gamma(x^\gamma).$$

Suppose the statement holds for $i \geq \ell$. When $i = \ell - 1$, we have

$$\Pr\left(\tau \geq \ell - 1 \mid X^{\ell-1} = x^{\ell-1}\right)$$

$$= \sum_{x_\ell \in \mathcal{X}} \mathcal{M}_s(x_\ell \mid \boldsymbol{c}, x^{\ell-1}) \cdot \Pr\left(\tau \geq \ell - 1 \mid X^\ell = x^\ell\right)$$

$$= \sum_{x_\ell \in \mathcal{X}} \mathcal{M}_s(x_\ell \mid \boldsymbol{c}, x^{\ell-1}) \cdot \left(\Pr\left(\tau \geq \ell \mid X^\ell = x^\ell\right) + \Pr\left(\tau = \ell - 1 \mid X^\ell = x^\ell\right)\right)$$

$$= \sum_{x_\ell \in \mathcal{X}} \mathcal{M}_s(x_\ell \mid \boldsymbol{c}, x^{\ell-1}) \cdot \left(\Pr\left(\tau \geq \ell \mid X^\ell = x^\ell\right) + \Pr\left(\tau < \ell \mid X^\ell = x^\ell\right) \cdot h_{\ell-1}^{\text{block}}\right) \quad (25)$$

$$= \sum_{x_\ell \in \mathcal{X}} \mathcal{M}_s(x_\ell \mid \boldsymbol{c}, x^{\ell-1}) \cdot \left(p_\ell(x^\ell) + (1 - p_\ell(x^\ell)) \cdot h_{\ell-1}^{\text{block}}\right),$$

$$= \sum_{x_\ell \in \mathcal{X}} \mathcal{M}_s(x_\ell \mid \boldsymbol{c}, x^{\ell-1}) \cdot p_\ell(x^\ell) + h_{\ell-1}^{\text{block}} \cdot \sum_{x_\ell \in \mathcal{X}} \mathcal{M}_s(x_\ell \mid \boldsymbol{c}, x^{\ell-1}) \cdot (1 - p_\ell(x^\ell))$$

$$= \sum_{x_\ell \in \mathcal{X}} \mathcal{M}_s(x_\ell \mid \boldsymbol{c}, x^{\ell-1}) \cdot p_\ell(x^\ell) + h_{\ell-1}^{\text{block}} \cdot (1 - \sum_{x_\ell \in \mathcal{X}} \mathcal{M}_s(x_\ell \mid \boldsymbol{c}, x^{\ell-1}) p_\ell(x^\ell)). \quad (26)$$

Equation (25) above holds since $\tau = \ell - 1$ happens under the joint event of $\tau < \ell$ and $\eta_{\ell-1} < h_{\ell-1}^{\text{block}}$. Note that in the definition of $h_{\ell-1}^{\text{block}}$ (Equation (5)),

$$\sum_x \max\{p_{\ell-1}(x^{\ell-1})\mathcal{M}_b(x \mid \boldsymbol{c}, x^{\ell-1}) - \mathcal{M}_s(x \mid \boldsymbol{c}, x^{\ell-1}), 0\}$$

$$= \sum_x \left(p_{\ell-1}(x^{\ell-1})\mathcal{M}_b(x \mid \boldsymbol{c}, x^{\ell-1}) - \min\{p_{\ell-1}(x^{\ell-1})\mathcal{M}_b(x \mid \boldsymbol{c}, x^{\ell-1}), \mathcal{M}_s(x \mid \boldsymbol{c}, x^{\ell-1})\}\right)$$

$$= p_{\ell-1}(x^{\ell-1}) - \sum_x \min\{p_{\ell-1}(x^{\ell-1})\mathcal{M}_b(x \mid \boldsymbol{c}, x^{\ell-1}), \mathcal{M}_s(x \mid \boldsymbol{c}, x^{\ell-1})\}$$

Plugging this into Equation (5),

$$h_{\ell-1}^{\text{block}} = \frac{p_{\ell-1}(x^{\ell-1}) - \sum_x \min\{p_{\ell-1}(x^{\ell-1})\mathcal{M}_b(x \mid \boldsymbol{c}, x^{\ell-1}), \mathcal{M}_s(x \mid \boldsymbol{c}, x^{\ell-1})\}}{1 - \sum_x \min\{p_{\ell-1}(x^{\ell-1})\mathcal{M}_b(x \mid \boldsymbol{c}, x^{\ell-1}), \mathcal{M}_s(x \mid \boldsymbol{c}, x^{\ell-1})\}}. \quad (27)$$

Moreover, we have by the definition of $p_\ell(x^\ell)$,

$$\sum_{x_\ell \in \mathcal{X}} \mathcal{M}_s(x_\ell \mid \boldsymbol{c}, x^{\ell-1}) \cdot p_\ell(x^\ell) = \sum_{x_\ell \in \mathcal{X}} \min\{p_{\ell-1}(x^{\ell-1})\mathcal{M}_b(x_\ell \mid \boldsymbol{c}, x^{\ell-1}), \mathcal{M}_s(x_\ell \mid \boldsymbol{c}, x^{\ell-1})\}$$

$$= \sum_{x \in \mathcal{X}} \min\{p_{\ell-1}(x^{\ell-1})\mathcal{M}_b(x \mid \boldsymbol{c}, x^{\ell-1}), \mathcal{M}_s(x \mid \boldsymbol{c}, x^{\ell-1})\}. \quad (28)$$

Plugging Equation (28) and Equation (27) into Equation (26), we get

$$\Pr\left(\tau \geq \ell - 1 \mid X^{\ell-1} = x^{\ell-1}\right) = p_{\ell-1}(x^{\ell-1}),$$

as desired. The lemma hence follows by induction.

## B.4 PROOF OF LEMMA 4

Recall that we use $O^{\gamma+1}$ to denote the sequence $(X^\tau, Y, Z^{\gamma-\tau})$ in Equation (7). Without loss of generality, we only consider $o^\ell$ such that $\Pr\left(O^\ell = o^\ell\right) > 0$ and $\Pr\left(X^\ell = o^\ell\right) > 0$ since otherwise $\Pr\left(\tau \geq i \mid X^i = x^i\right)$ is either zero or ill-defined. We break the proof into the two cases below.

If $\forall i < \ell$, it satisfies that $p_{i-1}(x^{i-1})\mathcal{M}_b(x_i \mid \boldsymbol{c}, x^{i-1}) \leq \mathcal{M}_s(x_i \mid \boldsymbol{c}, x^{i-1})$, then we have in the recursive formula of $p_i$'s in Algorithm 2, we always have

$$p_i(x^i) = p_{i-1}(x^{i-1}) \frac{\mathcal{M}_b(x_i \mid \boldsymbol{c}, x^{i-1})}{\mathcal{M}_s(x_i \mid \boldsymbol{c}, x^{i-1})},$$

and hence

$$p_{\ell-1}(x^{\ell-1}) = \frac{\mathcal{M}_b(x^{\ell-1} \mid \boldsymbol{c})}{\mathcal{M}_s(x^{\ell-1} \mid \boldsymbol{c})},$$

And for $x^\ell$, we have

$$p_\ell(x^\ell) = \min\{\frac{\mathcal{M}_b(x^\ell \mid \boldsymbol{c})}{\mathcal{M}_s(x^\ell \mid \boldsymbol{c})}, 1\}.$$

Note that

$$\Pr\left(O^\ell = x^\ell, \tau \geq \ell\right) = \Pr\left(X^\ell = x^\ell\right)\Pr\left(\tau \geq \ell \mid X^\ell = x^\ell\right) = \mathcal{M}_s(x^\ell \mid \boldsymbol{c})\Pr\left(\tau \geq \ell \mid X^\ell = x^\ell\right)$$

Moreover, we have

$$\Pr\left(O^\ell = x^\ell, \tau \geq \ell\right) \leq \Pr\left(O^\ell = x^\ell\right) = \mathcal{M}_b(x^\ell \mid \boldsymbol{c}),$$

and

$$\Pr\left(O^\ell = x^\ell, \tau \geq \ell\right) = \Pr\left(X^\ell = x^\ell\right)\Pr\left(\tau \geq \ell \mid X^\ell = x^\ell\right) \leq \Pr\left(X^\ell = x^\ell\right) = \mathcal{M}_s(x^\ell \mid \boldsymbol{c}).$$

Hence

$$\Pr\left(\tau \geq \ell \mid X^\ell = x^\ell\right) = \frac{\Pr\left(O^\ell = x^\ell, \tau \geq \ell\right)}{\mathcal{M}_s(x^\ell \mid \boldsymbol{c})} \leq \min\{\frac{\mathcal{M}_b(x^\ell \mid \boldsymbol{c})}{\mathcal{M}_s(x^\ell \mid \boldsymbol{c})}, 1\} = p_\ell(x^\ell).$$

In the other case, there must exist some $i$ such that $p_{i-1}(x^{i-1})\mathcal{M}_b(x_i \mid \boldsymbol{c}, x^{i-1}) > \mathcal{M}_s(x_i \mid \boldsymbol{c}, x^{i-1})$, then we have

$$p_i(x^i) = \min\{\frac{p_{i-1}(x^{i-1})\mathcal{M}_b(x_i \mid \boldsymbol{c}, x^{i-1})}{\mathcal{M}_s(x_i \mid \boldsymbol{c}, x^{i-1})}, 1\} = 1.$$

WLOG, let $i$ be the largest such index. In this case, we have $\forall i < j < \ell, p_{j-1}(x^{j-1})\mathcal{M}_b(x_j \mid \boldsymbol{c}, x^{j-1}) \leq \mathcal{M}_s(x_j \mid \boldsymbol{c}, x^{j-1})$, and hence

$$p_\ell(x^\ell) = p_i(x^i)\frac{\mathcal{M}_b^{\ell-i}(x^{i+1:\ell} \mid \boldsymbol{c}, x^i)}{\mathcal{M}_s^{\ell-i}(x^{i+1:\ell} \mid \boldsymbol{c}, x^i)} = \frac{\mathcal{M}_b^{\ell-i}(x^{i+1:\ell} \mid \boldsymbol{c}, x^i)}{\mathcal{M}_s^{\ell-i}(x^{i+1:\ell} \mid \boldsymbol{c}, x^i)}.$$

Moreover, by definition, we have

$$p_{i-1}(x^{i-1}) \leq p_{i-2}(x^{i-2})\frac{\mathcal{M}_b(x_{i-1} \mid \boldsymbol{c}, x^{i-2})}{\mathcal{M}_s(x_{i-1} \mid \boldsymbol{c}, x^{i-2})} \leq \cdots \leq \frac{\mathcal{M}_b^{i-1}(x^{i-1} \mid \boldsymbol{c})}{\mathcal{M}_s^{i-1}(x^{i-1} \mid \boldsymbol{c})},$$

and hence when $p_{i-1}(x^{i-1})\mathcal{M}_b(x_i \mid \boldsymbol{c}, x^{i-1}) > \mathcal{M}_s(x_i \mid \boldsymbol{c}, x^{i-1})$,

$$\frac{\mathcal{M}_b(x^i \mid \boldsymbol{c})}{\mathcal{M}_s(x^i \mid \boldsymbol{c})} = \frac{\mathcal{M}_b^{i-1}(x^{i-1} \mid \boldsymbol{c})}{\mathcal{M}_s^{i-1}(x^{i-1} \mid \boldsymbol{c})} \cdot \frac{\mathcal{M}_b(x_i \mid \boldsymbol{c}, x^{i-1})}{\mathcal{M}_s(x_i \mid \boldsymbol{c}, x^{i-1})} \geq p_{i-1}(x^{i-1}) \cdot \frac{\mathcal{M}_b(x_i \mid \boldsymbol{c}, x^{i-1})}{\mathcal{M}_s(x_i \mid \boldsymbol{c}, x^{i-1})} > 1.$$

Hence

$$\Pr\left(O^i = x^i, \tau \geq i\right) = \Pr\left(X^i = x^i\right)\Pr\left(\tau \geq i \mid X^i = x^i\right) \leq \mathcal{M}_s(x^i \mid \boldsymbol{c}) < \mathcal{M}_b(x^i \mid \boldsymbol{c}),$$

and

$$\Pr\left(O^i = x^i, \tau < i\right) = \Pr\left(O^i = x^i\right) - \Pr\left(O^i = x^i, \tau \geq i\right) = \mathcal{M}_b(x^i \mid \boldsymbol{c}) - \mathcal{M}_s(x^i \mid \boldsymbol{c}) > 0.$$

Note that when $O^i = x^i, \tau < i$, by constraints in Equation (7), we have

$$\Pr\left(O^{i+1:\ell} = x^{i+1:\ell} \mid O^i = x^i, \tau < i\right) = \mathcal{M}_b^{\ell-i}(x^{i+1:\ell} \mid \boldsymbol{c}, x^i).$$

This implies

$$\begin{aligned}\Pr\left(O^\ell = x^\ell\right) &= \Pr\left(O^i = x^i, \tau < i\right) \cdot \Pr\left(O^{i+1:\ell} = x^{i+1:\ell} \mid O^i = x^i, \tau < i\right) \\ &\quad + \Pr\left(O^i = x^i, \tau \geq i\right)\Pr\left(O^{i+1:\ell} = x^{i+1:\ell} \mid O^i = x^i, \tau \geq i\right) \\ &= \Pr\left(O^i = x^i, \tau < i\right) \cdot \mathcal{M}_b^{\ell-i}(x^{i+1:\ell} \mid \boldsymbol{c}, x^i) \\ &\quad + \Pr\left(O^i = x^i, \tau \geq i\right)\Pr\left(O^{i+1:\ell} = x^{i+1:\ell} \mid O^i = x^i, \tau \geq i\right)\end{aligned}$$

Moreover, we have

$$\Pr\left(O^\ell = x^\ell\right) = \mathcal{M}_b(x^i)\mathcal{M}_b^{\ell-i}(x^{i+1:\ell} \mid \boldsymbol{c}, x^i)$$

Combining both, we get

$$
\begin{aligned}
1 &= \frac{\Pr\left(O^\ell = x^\ell\right)}{\mathcal{M}_b(x^i)\mathcal{M}_b^{\ell-i}(x^{i+1:\ell} \mid \boldsymbol{c}, x^i)} \\
&= \Pr\left(\tau < i \mid O^i = x^i\right) + \Pr\left(\tau \geq i \mid O^i = x^i\right)\frac{\Pr\left(O^{i+1:\ell} = x^{i+1:\ell} \mid O^i = x^i, \tau \geq i\right)}{\mathcal{M}_b^{\ell-i}(x^{i+1:\ell} \mid \boldsymbol{c}, x^i)}, \\
&= 1 - \Pr\left(\tau \geq i \mid O^i = x^i\right)\left(\frac{\Pr\left(O^{i+1:\ell} = x^{i+1:\ell} \mid O^i = x^i, \tau \geq i\right)}{\mathcal{M}_b^{\ell-i}(x^{i+1:\ell} \mid \boldsymbol{c}, x^i)} - 1\right),
\end{aligned}
$$

and this implies that (note by assumption $\Pr\left(\tau \geq i \mid O^i = x^i\right) \neq 0$),

$$\Pr\left(O^{i+1:\ell} = x^{i+1:\ell} \mid O^i = x^i, \tau \geq i\right) = \mathcal{M}_b^{\ell-i}(x^{i+1:\ell} \mid \boldsymbol{c}, x^i). \tag{29}$$

Hence

$$
\begin{aligned}
\Pr\left(O^\ell = x^\ell, \tau \geq \ell\right) &\leq \Pr\left(O^\ell = x^\ell, \tau \geq i\right) \\
&= \Pr\left(O^i = x^i, \tau \geq i\right)\Pr\left(O^{i+1:\ell} = x^{i+1:\ell} \mid O^i = x^i, \tau \geq i\right) \\
&\leq \mathcal{M}_s(x^i \mid \boldsymbol{c})p_i(x^i)\mathcal{M}_b^{\ell-i}(x^{i+1:\ell} \mid \boldsymbol{c}, x^i) \\
&= \mathcal{M}_s(x^i \mid \boldsymbol{c})\mathcal{M}_b^{\ell-i}(x^{i+1:\ell} \mid \boldsymbol{c}, x^i).
\end{aligned}
$$

If $\Pr\left(\tau \geq \ell \mid X^\ell = x^\ell\right) > p_\ell(x^\ell)$, we have

$$
\begin{aligned}
\Pr\left(O^\ell = x^\ell, \tau \geq \ell\right) &= \Pr\left(X^\ell = x^\ell\right)\Pr\left(\tau \geq \ell \mid X^\ell = x^\ell\right) \\
&> \mathcal{M}_s(x^\ell \mid \boldsymbol{c})p_\ell(x^\ell) \\
&= \mathcal{M}_s(x^\ell \mid \boldsymbol{c})\frac{\mathcal{M}_b^{\ell-i}(x^{i+1:\ell} \mid \boldsymbol{c}, x^i)}{\mathcal{M}_s^{\ell-i}(x^{i+1:\ell} \mid \boldsymbol{c}, x^i)} \\
&= \mathcal{M}_s(x^i \mid \boldsymbol{c})\mathcal{M}_b^{\ell-i}(x^{i+1:\ell} \mid \boldsymbol{c}, x^i),
\end{aligned}
$$

which leads to a contradiction. This completes the proof.

## C   GREEDY BLOCK VERIFICATION

In this section, we show that it is possible to accept more tokens than block verification (Algorithm 2) *in one iteration* with a modification to the speculative decoding framework in Algorithm 3 that allows the decoding logic to depend on the previous accept/reject decisions. However, as shown in Table 4, the resulting algorithm, greedy block verification, doesn't improve over block verification. We include the description and analysis of the algorithm as a theoretical result. The claims in the main paper holds independent of the results in this section.

We start by introducing the algorithm (Algorithm 4) and then discuss the necessary modifications to maintain the identity distribution guarantee.

The above greedy block verification algorithm has a similar procedure as block verification (Algorithm 2) with differences in the setting of of acceptance probabilities and residual distributions, as highlighted.

Similar to Algorithm 2, Algorithm 4 maintains a list of probabilities $\tilde{p}_i$'s, which satisfies that $\min\{1, \tilde{p}_i\}$ is the probability that the subblock $X^i$ is accepted. $h_i$'s are chosen to achieve the above acceptance guarantee, and $p_{\text{res}}^{\text{greedy}}$'s are chosen to maintain the identical distribution guarantee.

Note that compared to $p_i$ in block verification, the recursive definition of $\tilde{p}_i$ doesn't have a minimum over one term, hence it is always an upper bound on $p_i$'s. This leads to a higher acceptance probability for every subblock in greedy block verification (Theorem 3). However, Algorithm 4 cannot be used directly in the iterative implementation of speculative decoding in Algorithm 3. To see this, consider the simple example in Section 2. Greedy block verification will perform the following:

---

**Algorithm 4** Greedy block verification

---

**Input:** Draft block $X^\gamma$; small model distributions $\forall i < \gamma, \mathcal{M}_s(\cdot \mid \boldsymbol{c}, X^i)$; target model distributions $\forall i \leq \gamma, \mathcal{M}_b(\cdot \mid \boldsymbol{c}, X^i)$.

1: Sample $\eta_1, \ldots, \eta_\gamma \sim U(0, 1)$.
2: Set $\tau = 0$, $p_0 = 1$.
3: **for** $i = 1, \ldots,$ $\gamma - 1$ **do**

4:  Set $\tilde{p}_i = \tilde{p}_{i-1} \frac{\mathcal{M}_b(X_i \mid \boldsymbol{c}, X^{i-1})}{\mathcal{M}_s(X_i \mid \boldsymbol{c}, X^{i-1})}$.

5:  Set $h_i = \frac{\sum_x \max\{\tilde{p}_i \mathcal{M}_b(x \mid \boldsymbol{c}, X^i) - \mathcal{M}_s(x \mid \boldsymbol{c}, X^i), 0\}}{\sum_x \max\{\mathcal{M}_s(x \mid \boldsymbol{c}, X^i) - \tilde{p}_i \mathcal{M}_b(x \mid \boldsymbol{c}, X^i), 0\}}$

6:  **if** $\eta_i \leq h_i$ **then**
7:    Set $\tau = i$.
8:  **else**
9:    **continue.**
10:  **end if**
11: **end for**

12: $\tilde{p}_\gamma = \tilde{p}_{\gamma-1} \frac{\mathcal{M}_b(X_\gamma \mid \boldsymbol{c}, X^{\gamma-1})}{\mathcal{M}_s(X_\gamma \mid \boldsymbol{c}, X^{\gamma-1})}$

13: **if** $\eta_\gamma < \tilde{p}_\gamma$ **then**
14:  Set $\tau = \gamma$, and sample $Y$ from $\mathcal{M}_b(\cdot \mid \boldsymbol{c}, X^\gamma)$.
15: **else**
16:  Sample $Y$ from $p_{\text{res}}^{\text{greedy}}(\cdot \mid \boldsymbol{c}, X^\tau)$ as below:

$$p_{\text{res}}^{\text{greedy}}(x \mid \boldsymbol{c}, X^i) = \frac{\max\{\tilde{p}_i \cdot \mathcal{M}_b(x \mid \boldsymbol{c}, X^i) - \mathcal{M}_s(x \mid \boldsymbol{c}, X^i), 0\}}{\sum_{x' \in \mathcal{X}} \max\{\tilde{p}_i \cdot \mathcal{M}_b(x' \mid \boldsymbol{c}, X^i) - \mathcal{M}_s(x' \mid \boldsymbol{c}, X^i), 0\}}. \quad (30)$$

17: **end if**
18: **Return** $X^\tau, Y$.

---

Accept $X_1 X_2 = \text{AB}, \text{BA}, \text{BB}$ with probability one, and sample an extra token from $\mathcal{M}_b(\cdot)$. Accept $X_1 X_2 = \text{AA}$ with probability $1/4$ and sample an extra token from $\mathcal{M}_b(\cdot)$. When $X_1 X_2 = \text{AA}$ is rejected, accept no tokens and sample a correction token $Y = \text{B}$. Note that in this case, if the algorithm uses $Y$ as the context for the next iteration and sample based on $\mathcal{M}_b$, the next token will be A with probability $1/3$. This makes the total probability of generating BA as the first two tokens

$$\mathcal{M}_s(\text{BA}) \Pr(\text{Accept BA}) + \mathcal{M}_s(\text{AA}) \Pr(\text{Reject AA and } Y = \text{B}) \mathcal{M}_b(\text{A}) = 2/9 \cdot 1 + 4/9 \cdot 3/4 \cdot 1/3$$
$$= 1/3,$$

which is higher than $\mathcal{M}_b(\text{BA}) = 2/9$. This violates the identical distribution guarantee. Below we introduce a distribution modification algorithm, which can be used with Algorithm 4 to maintain the identical distribution guarantee.

It can be shown that if $X^\tau, Y$ are returned in Algorithm 4, and the next $\tau - \gamma - 1$ tokens are sampled according to $\mathcal{M}_{\text{new}}$ from Algorithm 5, the identical distribution guarantee is maintained. In particular, we have the following lemma:

**Lemma 8.** *Let $X^\gamma \sim \mathcal{M}_s^\gamma(\cdot \mid \boldsymbol{c})$ be the draft tokens and $X^\tau, Y$ be the output from Algorithm 4. Let $\mathcal{M}_{\text{new}}$ be the modified distribution based on Algorithm 5, and $Z^{\gamma-\tau-1} \sim \mathcal{M}_{\text{new}}^{\gamma-\tau-1}(\cdot \mid \boldsymbol{c}, X^\tau, Y)$. Then we have*

$$X^\tau, Y, Z^{\gamma-\tau-1} \sim \mathcal{M}_b^\gamma(\cdot \mid \boldsymbol{c}).$$

The proof is presented in Appendix C.3. The above leads to the following speculative decoding algorithm with greedy block verification, presented in Algorithm 6. Note that Lemma 8 implies that it maintains the identical distribution guarantee.

---

**Algorithm 5** Distribution modification

---

**Input:** Small model $\mathcal{M}_s$; target model $\mathcal{M}_b$; draft length $\gamma$; generated tokens from Algorithm 4 $X^\tau, Y$.

1: Let $\mathcal{M}'_b$ be such that $\forall i \leq \gamma - \tau - 1$, and $x^i \in \mathcal{X}^i$, $\mathcal{M}_{\text{new}}(x_i \mid \boldsymbol{c}, X^\tau, Y, x^{i-1}) =$

$$\frac{\max\{\mathcal{M}_b(\boldsymbol{c}, X^\tau, Y, x^i) - \mathcal{M}_s(\boldsymbol{c}, X^\tau, Y, x^i), 0\}}{\sum_{x' \in \mathcal{X}} \max\{\mathcal{M}_b(\boldsymbol{c}, X^\tau, Y, x^{i-1}, x') - \mathcal{M}_s(\boldsymbol{c}, X^\tau, Y, x^{i-1}, x'), 0\}}, \quad (31)$$

{Modify the distribution at rejected locations.}

and $\forall i > \gamma - \tau - 1$, and $x^i \in \mathcal{X}^i$,

$$\mathcal{M}_{\text{new}}(x_i \mid \boldsymbol{c}, X^\tau, Y, x^{i-1}) = \mathcal{M}_b(x_i \mid \boldsymbol{c}, X^\tau, Y, x^{i-1})$$

{Keep the distributions for future locations unchanged.}

2: **Return** $\mathcal{M}_{\text{new}}$.

---

**Algorithm 6** Speculative decoding with greedy block verification

---

**Input:** Prefix $\boldsymbol{c}$, large model $\mathcal{M}_b$, draft model $\mathcal{M}_s$. Draft length $\gamma$..

1: **while** E.O.S $\notin (X^\tau, Y)$ **do**
2:  Sample $X_1, \ldots, X_\gamma \sim \mathcal{M}_s(\cdot \mid \boldsymbol{c})$ using autoregressive sampling, keep the conditional probabilities at each step $\mathcal{M}_s(\cdot \mid \boldsymbol{c}, X^i)$ for $i = 0, \ldots, \gamma - 1$. {Obtain draft block.}
3:  Call the large model $\mathcal{M}_b$ and compute conditional probabilities $\mathcal{M}_b(\cdot \mid \boldsymbol{c}, X^i)^6$ for $i = 0, 1, \ldots, \gamma$ in parallel. {Parallel scoring.}
4:  Get the accepted tokens with draft verification {Draft verification and correction.}

$$X^\tau, Y = \text{VERIFY}(X^\gamma, \{\mathcal{M}_s(\cdot \mid \boldsymbol{c}, X^i)\}_{i=0}^{\gamma-1}, \{\mathcal{M}_b(\cdot \mid \boldsymbol{c}, X^i)\}_{i=0}^{\gamma}).$$

5:  $\boldsymbol{c} \leftarrow \boldsymbol{c}, X^\tau, Y$. {Add decoded tokens to the prefix.}
6:  $\mathcal{M}_b \leftarrow \text{DISTRIBUTIONMODIFY}(\mathcal{M}_b, \mathcal{M}_s, \gamma, X^\tau, Y)$ {Modify target distribution.}
7: **end while**

---

## C.1 COMPARISON TO BLOCK VERIFICATION.

In one draft iteration, with the same pair of draft and target distributions, greedy block verification is always better.

**Theorem 3** (Informal). *In one draft iteration with the same models $\mathcal{M}_s, \mathcal{M}_b$ and draft length $\gamma$, greedy block verification decodes at least as many tokens as block verification.*

The theorem is proved in Appendix C.3. However, due to the distribution modification step, the target distribution might change after the first iteration, which might affect the expected number of accepted tokens. For example, in the Bernoulli example considered in Section 2, when the draft block $X_1 X_2 = \text{AA}$ and they are rejected by greedy block verification. It can be shown that the modified distribution will be a point mass on token B. And in future iterations, if the algorithm still uses $\mathcal{M}_s$ as the draft model, there is lower chance that the draft tokens will be accepted. Hence, theoretically it is unclear whether one approach dominates the other.

**Empirical comparison.** We conduct the same set of experiments in Section 6 on greedy block verification to compare the two approaches empirically. We list the block efficiency comparison when PALM-2-XXS is used as the drafter and $\gamma = 8$ in Table 4. As we can see, while greedy block verification still consistently improves over token verification, the improvement is less significant compared to block verification. The trend is the same for wall clock numbers as well as in other parameter settings. Hence we recommend using block verification instead of the greedy version.

---

[6]In cases where $\mathcal{M}_b$ is not the original large transformer model. $\mathcal{M}_b$ can be obtained by evaluating using the original large model, and then perform the modification in Equation (31).

Table 4: Block efficiency comparison between among token verification, block verification, and greedy block verification with $\gamma = 8$. Each statistic is computed using 1000 test prompts from different datasets on various tasks (each run is an average with 3 different random seeds).

| Dataset | Token Verification | Block verification | Greedy block verification |
|---|---|---|---|
| LM1B | 3.21 | **3.49** | 3.30 |
| GPT Prompt | 3.41 | **3.76** | 3.51 |
| WebQA | 3.44 | **3.70** | 3.52 |
| PIQA | 3.40 | **3.68** | 3.49 |
| ShareGPT | 3.34 | **3.62** | 3.44 |
| XSum | 3.49 | **3.76** | 3.59 |
| GSM8K | 3.81 | **4.15** | 3.96 |
| WMT-DeEn | 3.19 | **3.41** | 3.26 |

### C.2 PROOF OF LEMMA 8

In the proof, we ignore the context $c$ and the proof will generalize to arbitrary $c$. We start by introducing two useful quantities.

$$p_{\text{remain}}(x^i) := \sum_x \max\{\mathcal{M}_b(x^i, x) - \mathcal{M}_s(x^i, x), 0\}, \tag{32}$$

$$p_{\text{rej}}(x^i) := \sum_x \max\{\mathcal{M}_s(x^i, x) - \mathcal{M}_b(x^i, x), 0\}. \tag{33}$$

Note that $p_i$ in Algorithm 4 depends on the draft block $x^i$, and by the recursive definition of $\tilde{p}_i$'s, we have $\tilde{p}_i = \frac{\mathcal{M}_b(x^i)}{\mathcal{M}_s(x^i)}$. Hence we have

$$h_i = \frac{\sum_x \max\{\tilde{p}_i \mathcal{M}_b(x \mid c, X^i) - \mathcal{M}_s(x \mid c, X^i), 0\}}{\sum_x \max\{\mathcal{M}_s(x \mid c, X^i) - \tilde{p}_i \mathcal{M}_b(x \mid c, X^i), 0\}} = \frac{p_{\text{remain}}(x^i)}{p_{\text{rej}}(x^i)}.$$

Moreover, the expression for $\tilde{p}_i$ also implies that $p_{\text{res}}^{\text{greedy}}(\cdot \mid c, X^\tau) = \mathcal{M}_{\text{new}}(\cdot \mid c, X^\tau)$ (defined in Equation (30) and Equation (31) resepectively).

We now prove the following lemma about the acceptance length $\tau$ in Algorithm 4.

**Lemma 9.** *For all $\ell \in [1, \gamma]$, and $x^\ell \in \mathcal{X}^\ell$,*

$$\Pr\left(X^\ell = x^\ell, \tau \geq \ell\right) = \min\{\mathcal{M}_b(x^\ell), \mathcal{M}_s(x^\ell)\}.$$

*Proof.* We prove this by induction in the backward direction. When $\ell = \gamma$, Step 12-14 in Algorithm 4 accepts $X^\gamma$ with probability

$$\Pr\left(\tau = \gamma \mid X^\gamma = x^\gamma\right) = \min\{1, \tilde{p}_\gamma\} = \min\left\{1, \frac{\mathcal{M}_b(x^\gamma)}{\mathcal{M}_s(x^\gamma)}\right\},$$

and hence

$$\Pr\left(X^\gamma = x^\gamma, \tau \geq \gamma\right) = \Pr\left(X^\gamma = x^\gamma\right)\Pr\left(\tau = \gamma \mid X^\gamma = x^\gamma\right) = \min\left\{\mathcal{M}_s(x^\gamma), \mathcal{M}_b(x^\gamma)\right\}.$$

Suppose the equation holds for $\ell \geq \ell_0$, for $\ell = \ell_0 - 1$, we have

$$\Pr\left(X^{\ell_0-1} = x^{\ell_0-1}, \tau \geq \ell_0 - 1\right) = \Pr\left(X^{\ell_0-1} = x^{\ell_0-1}, \tau \geq \ell_0\right) + \Pr\left(X^{\ell_0-1} = x^{\ell_0-1}, \tau = \ell_0 - 1\right).$$

Next we consider the two terms separately. For the first term, due to the induction assumption, we have

$$\Pr\left(X^{\ell_0-1} = x^{\ell_0-1}, \tau \geq \ell_0\right) = \sum_{x \in \mathcal{X}} \min\{\mathcal{M}_b(x^{\ell_0-1}, x), \mathcal{M}_s(x^{\ell_0-1}, x)\}$$

For the second term, we have

$$
\begin{aligned}
&\Pr\left(X^{\ell_0-1} = x^{\ell_0-1}, \tau = \ell_0 - 1\right) \\
&= \Pr\left(X^{\ell_0-1} = x^{\ell_0-1}, \tau \le \ell_0 - 1\right) \cdot \Pr\left(X^{\ell_0-1} \text{ is accepted.}\right) \quad\quad (34)\\
&= \Pr\left(X^{\ell_0-1} = x^{\ell_0-1}, \tau \le \ell_0 - 1\right) \cdot \Pr\left(\eta_{\ell_0-1} \le h_{\ell_0-1}\right) \\
&= \left(\Pr\left(X^{\ell_0-1} = x^{\ell_0-1}\right) - \Pr\left(X^{\ell_0-1} = x^{\ell_0-1}, \tau \ge \ell_0\right)\right) \cdot \min\left\{1, \frac{p_{\text{remain}}(x^{\ell_0-1})}{p_{\text{rej}}(x^{\ell_0-1})}\right\} \\
&= \sum_{x \in \mathcal{X}} \left(\mathcal{M}_s(x^{\ell_0-1}, x) - \min\{\mathcal{M}_b(x^{\ell_0-1}, x), \mathcal{M}_s(x^{\ell_0-1}, x)\}\right) \cdot \min\left\{1, \frac{p_{\text{remain}}(x^{\ell_0-1})}{p_{\text{rej}}(x^{\ell_0-1})}\right\}
\end{aligned}
$$
$$(35)$$
$$
\begin{aligned}
&= p_{\text{rej}}(x^{\ell_0-1}) \cdot \min\left\{1, \frac{p_{\text{remain}}(x^{\ell_0-1})}{p_{\text{rej}}(x^{\ell_0-1})}\right\} \quad\quad\quad\quad\quad\quad\quad\quad\quad\quad\quad (36)\\
&= \min\left\{p_{\text{remain}}(x^{\ell_0-1}), p_{\text{rej}}(x^{\ell_0-1})\right\}.
\end{aligned}
$$

In the above derivation, Equation (34) is due to that Algorithm 4 is outputting the longest accepted subblock. Equation (35) is due to the induction hypothesis. Equation (36) is due to the definition of $p_{\text{rej}}$ in Equation (33).

Combining the two terms, we have

$$
\begin{aligned}
&\Pr\left(X^{\ell_0-1} = x^{\ell_0-1}, \tau \ge \ell_0 - 1\right) \\
&= \Pr\left(X^{\ell_0-1} = x^{\ell_0-1}, \tau \ge \ell_0\right) + \Pr\left(X^{\ell_0-1} = x^{\ell_0-1}, \tau = \ell_0 - 1\right) \\
&= \sum_{x \in \mathcal{X}} \min\{\mathcal{M}_b(x^{\ell_0-1}, x), \mathcal{M}_s(x^{\ell_0-1}, x)\} + \min\left\{p_{\text{remain}}(x^{\ell_0-1}), p_{\text{rej}}(x^{\ell_0-1})\right\} \\
&= \min\{\sum_{x \in \mathcal{X}} \min\{\mathcal{M}_b(x^{\ell_0-1}, x), \mathcal{M}_s(x^{\ell_0-1}, x)\} + p_{\text{remain}}(x^{\ell_0-1}), \\
&\qquad\qquad \sum_{x \in \mathcal{X}} \min\{\mathcal{M}_b(x^{\ell_0-1}, x), \mathcal{M}_s(x^{\ell_0-1}, x)\} + p_{\text{rej}}(x^{\ell_0-1})\} \\
&= \min\left\{\sum_{x \in \mathcal{X}} \mathcal{M}_b(x^{\ell_0-1}, x), \sum_{x \in \mathcal{X}} \mathcal{M}_s(x^{\ell_0-1}, x)\right\} \quad\quad\quad\quad\quad\quad (37)\\
&= \min\left\{\mathcal{M}_b(x^{\ell_0-1}), \mathcal{M}_s(x^{\ell_0-1})\right\},
\end{aligned}
$$

which is the desired quantity in the lemma. This concludes the proof. Here Equation (37) is due to the definition of $p_{\text{remain}}$ and $p_{\text{rej}}$ in Equation (32) and Equation (33). $\qquad\square$

Next we proceed to prove Lemma 8. Let $O^\gamma = (X^\tau, Y, Z^{\gamma-\tau-1})$. It would be enough to show that for all $i \le \gamma$ and $x^i \in \mathcal{X}^i$,

$$
\Pr\left(O^i = x^i\right) = \mathcal{M}_b(x^i).
$$

We prove this via induction. Note that the corollary holds for $i = 0$, which is the trivial case and both sides are equal to 1. Suppose the claim holds for $i \le i_0$. This means $\forall x^{i_0} \in \mathcal{X}^{i_0}$, we have

$$
\Pr\left(O^{i_0} = x^{i_0}\right) = \mathcal{M}_b(x^{i_0}).
$$

When $i = i_0 + 1$, by the algorithm, we have that either $\tau \ge i_0 + 1$, where $O_{i_0+1}$ is an accepted token, or $\tau \le i_o$, where $O_{i_0+1}$ is sampled according to $\mathcal{M}_{\text{new}}(\cdot \mid O^{i_0})$ (or $p_{\text{res}}^{\text{greedy}}(\cdot \mid O^{i_0})$, which

is the same as $\mathcal{M}_{\text{new}}(\cdot \mid O^{i_0}))$. By Lemma 9, we have

$$
\begin{aligned}
&\Pr\left(O^{i_0+1} = x^{i_0+1}\right) \\
&= \Pr\left(O^{i_0+1} = x^{i_0+1}, \tau \geq i_0 + 1\right) + \Pr\left(O^{i_0+1} = x^{i_0+1}, \tau \leq i_0\right) \\
&= \Pr\left(X^{i_0+1} = x^{i_0+1}, \tau \geq i_0 + 1\right) + \Pr\left(O^{i_0} = x^{i_0}, \tau \leq i_0\right) \cdot \mathcal{M}_{\text{new}}(x_{i_0+1} \mid x^{i_0}) \\
&= \Pr\left(X^{i_0+1} = x^{i_0+1}, \tau \geq i_0 + 1\right) \\
&\qquad\qquad + \left(\Pr\left(O^{i_0} = x^{i_0}\right) - \Pr\left(O^{i_0} = x^{i_0}, \tau \geq i_0 + 1\right)\right) \cdot \mathcal{M}_{\text{new}}(x_{i_0+1} \mid x^{i_0})) \\
&= \min\{\mathcal{M}_b(x^{i_0+1}), \mathcal{M}_s(x^{i_0+1})\} \\
&\qquad\qquad + \left(\mathcal{M}_b(x^{i_0}) - \sum_x \min\{\mathcal{M}_b(x^{i_0}, x), \mathcal{M}_s(x^{i_0}, x)\}\right) \cdot \mathcal{M}_{\text{new}}(x_{i_0+1} \mid x^{i_0}))
\end{aligned}
\tag{38}
$$

$$
= \min\{\mathcal{M}_b(x^{i_0+1}), \mathcal{M}_s(x^{i_0+1})\} + \sum_x \max\{\mathcal{M}_b(x^{i_0}, x) - \mathcal{M}_s(x^{i_0}, x), 0\} \cdot \mathcal{M}_{\text{new}}(x_{i_0+1} \mid x^{i_0}))
$$

$$
= \min\{\mathcal{M}_b(x^{i_0+1}), \mathcal{M}_s(x^{i_0+1})\} + \max\{\mathcal{M}_b(x^{i_0}, x_{i_0+1}) - \mathcal{M}_s(x^{i_0}, x_{i_0+1}), 0\}
\tag{39}
$$

$$
= \mathcal{M}_b(x^{i_0+1}).
$$

Here Equation (38) follows by the induction hypothesis, and Equation (39) follows by the definition of $\mathcal{M}_{\text{new}}$ in Equation (31). By induction, this concludes the proof.

### C.3 PROOF OF THEOREM 3

To prove Theorem 3, we first observe the following: For Algorithm 4, let $\tau$ be the number of accepted tokens, due to Lemma 9, we have

$$
\begin{aligned}
\mathbb{E}_{X^\gamma \sim \mathcal{M}_s^\gamma}[\tau] &= \sum_{\ell=1}^{\gamma} \Pr\left(\tau \geq \ell\right) = \sum_{\ell=1}^{\gamma} \sum_{x^\ell \in \mathcal{X}^\ell} \Pr\left(X^\ell = x^\ell, \tau \geq \ell\right) \\
&= \sum_{\ell=1}^{\gamma} \sum_{x^\ell \in \mathcal{X}^\ell} \min\{\mathcal{M}_s(x^\ell), \mathcal{M}_b(x^\ell)\}.
\end{aligned}
$$

Next we show that the above expected acceptance length is optimal for a family of draft verification algorithms that performs a coupling between sample blocks from the draft and target distributions. For all $x^\gamma, y^\gamma \in \mathcal{X}^\gamma$, let the *maximum common prefix length* be defined as

$$
\beta(x^\gamma, y^\gamma) := \max_{\ell \leq \gamma}\{\forall i \leq \ell, x_i = y_i\}.
$$

Formally, let $\pi$ be a joint distribution over $\mathcal{X}^\gamma \times \mathcal{X}^\gamma$, then Algorithm 4 solves the following optimization problem.

$$
\max_\pi \mathbb{E}_{X^\gamma, y^\gamma \sim \pi}\left[\beta(X^\gamma, y^\gamma)\right],
\tag{40}
$$

subject to constraints

$$
\sum_{y^\gamma} \pi(x^\gamma, y^\gamma) = \mathcal{M}_s(x^\gamma), \quad \forall x^\gamma \in \mathcal{X}^\gamma,
\tag{41}
$$

$$
\sum_{x^\gamma} \pi(x^\gamma, y^\gamma) = \mathcal{M}_b(y^\gamma), \quad \forall y^\gamma \in \mathcal{X}^\gamma.
\tag{42}
$$

In this above formulation, the marginal distributions satisfy $X^\gamma \sim \mathcal{M}_s^\gamma$ and $Y^\gamma \sim \mathcal{M}_b^\gamma$. And the maximum common prefix length refers to the number of accepted tokens in one iteration of speculative decoding. Note that the optimization problem can be viewed as an optimal transport problem (Villani et al., 2009) between distributions $\mathcal{M}_s^\gamma(\cdot)$ and $\mathcal{M}_b^\gamma(\cdot)$ with the cost function being $(\gamma - \beta(X^\gamma, Y^\gamma))$. The next lemma establishes the optimality of Algorithm 4 in solving this problem.

**Lemma 10.** *The solution to Equation* (40) *is upper bounded by*

$$\sum_{\tau=1}^{\gamma} \sum_{x^{\tau} \in \mathcal{X}^{\tau}} \min\{\mathcal{M}_s(x^{\ell}), \mathcal{M}_b(x^{\ell})\}$$

**Proof of Lemma 10:** For all $\pi$ that satisfies Equations (41) and (42) and $X^{\gamma}, Y^{\gamma} \sim \pi$, we have

$$
\begin{aligned}
\mathbb{E}_{X^{\gamma},Y^{\gamma}}[\beta(X^{\gamma},Y^{\gamma})] &\overset{(a)}{=} \sum_{\ell \leq \gamma} \mathrm{Pr}_{X^{\gamma},Y^{\gamma}}\left(\beta(X^{\gamma},Y^{\gamma}) \geq \ell\right) \\
&= \sum_{\ell \leq \gamma} \sum_{x^{\ell}} \mathrm{Pr}_{X^{\gamma},Y^{\gamma}}\left(X^{\ell} = Y^{\ell} = x^{\ell}, \beta(X^{\gamma},Y^{\gamma}) \geq \ell\right) \\
&\leq \sum_{\ell \leq \gamma} \sum_{x^{\ell}} \mathrm{Pr}_{X^{\gamma},Y^{\gamma}}\left(X^{\ell} = Y^{\ell} = x^{\ell}\right) \\
&\overset{(b)}{\leq} \sum_{\ell \leq \gamma} \sum_{x^{\ell}} \min\left\{\mathrm{Pr}\left(X^{\ell} = x^{\ell}\right), \mathrm{Pr}\left(Y^{\ell} = x^{\ell}\right)\right\} \\
&= \sum_{\ell \leq \gamma} \sum_{x^{\ell}} \min\{\mathcal{M}_s^{\ell}(x^{\ell}), \mathcal{M}_b^{\ell}(x^{\ell})\}.
\end{aligned}
$$

Here $(a)$ follows from the fact that for a positive integer random variable $\mathbb{E}[X] = \sum_i \mathrm{Pr}(X \geq i)$; $(b)$ follows from the fact that the joint probability is upper bounded by the minimum of the marginals. $\square$

Then Theorem 3 holds by noticing that block verification Algorithm 2 is also an instance of the coupling by setting $X^{\gamma}$ to be the draft tokens and $Y^{\gamma} = (X^{\tau}, Y', Z^{\gamma-\tau-1})$ where $(X^{\tau}, Y')$ are the outputs from Algorithm 2 and $Z^{\gamma-\tau-1} \sim \mathcal{M}_b(\cdot \mid c, X^{\tau}, Y)$ (Lemma 2).

# D  ADDITIONAL EXPERIMENTAL RESULTS

## D.1  COMPARISON TO SPECULATIVE DECODING WITH MULTIPLE DRAFTS.

Recent works (Sun et al., 2023; Miao et al., 2023) have extended speculative decoding to the case with multiple draft blocks to improve block efficiency. However, these methods also increase the required computation from the large model to verify the drafts. In high-throughput LLM serving systems, query batching (Kwon et al., 2023) is a common technique where multiple prefixes are decoded at the same time. In these cases, the inference will be less memory bound and there will not be enough extra parallel compute to evaluate the increased number of drafts without decreasing latency.

We empirically compare block verification and SpecTr (Sun et al., 2023), SpecInfer (Miao et al., 2023) with query batching. We set the batch size $B = 8$ and use PALM-2-XXS as the draft model. In Table 5, we list the wall clock speedup and block efficiency for $\gamma = 8$. The number of draft blocks for SpecTr and SpecInfer are taken to be 2, which is the one that achieves the lowest latency over $\{2, 4, 8\}$ when $B = 8, \gamma = 8$.

We observe that while SpecTr and SpecInfer can achieve higher block efficiencies, due to the increased computation to evaluate more candidates, our method achieves better speedup than SpecTr and SpecInfer, demonstrating the advantage of our method in the common practical setting with query batching.

## D.2  DETAILED RESULTS WITH OTHER PARAMETER SETTINGS

In this section, we present experimental results for the same set of experiments described in Section 6 with different block lengths ($\gamma = 4, 6, 8$) and different drafters (PALM-2-XXS and PALM-2-XXXS):

- Table 6. Drafter: PALM-2-XXS, $\gamma = 4$.
- Table 7. Drafter: PALM-2-XXS, $\gamma = 6$.

Table 5: $\gamma = 8$, $B = 8$. Speedup comparison between token verification (TOKENV) and block verification (BLOCKV) with PALM-2-XXS as the draft model on various datasets and tasks.

| Dataset | Wall clock time over baseline | | | | Block efficiency | | | |
|---|---|---|---|---|---|---|---|---|
| | TOKENV | BLOCKV | SpecTr | SpecInfer | TOKENV | BLOCKV | SpecTr | SpecInfer |
| GPT Prompt | 1.300 | **1.381** | 1.290 | 1.263 | 3.394 | 3.715 | **3.898** | 3.833 |
| WebQA | 1.302 | **1.368** | 1.279 | 1.274 | 3.451 | 3.7 | **3.933** | 3.894 |
| ShareGPT | 1.267 | **1.333** | 1.244 | 1.236 | 3.366 | 3.63 | **3.824** | 3.78 |
| GSM8K | 1.353 | **1.445** | 1.344 | 1.319 | 3.856 | 4.179 | **4.356** | 4.277 |
| XSum | 1.328 | **1.403** | 1.300 | 1.285 | 3.487 | 3.768 | **3.949** | 3.897 |
| PIQA | 1.305 | **1.377** | 1.270 | 1.280 | 3.401 | 3.685 | **3.846** | 3.82 |
| LM1B | 1.274 | **1.344** | 1.253 | 1.245 | 3.218 | 3.494 | **3.669** | 3.629 |
| WMT-DeEn | 1.222 | **1.293** | 1.204 | 1.194 | 3.165 | 3.422 | **3.603** | 3.56 |

- Table 8. Drafter: PALM-2-XXXS, $\gamma = 4$.
- Table 9. Drafter: PALM-2-XXXS, $\gamma = 6$.
- Table 10. Drafter: PALM-2-XXXS, $\gamma = 8$.

Table 6: Speedup comparison between token verification (TOKENV) and block verification (BLOCKV) with $\gamma = 4$ and PALM-2-XXS being the draft model. Each statistic is computed using 1000 test prompts from different datasets on various tasks (each run is an average with 3 different random seeds). Numbers after $\pm$ represent standard deviation.

| Dataset | Block efficiency | | | Wall clock time speedup over baseline | | |
|---|---|---|---|---|---|---|
| | TOKENV | BLOCKV | Improve. ↑% | TOKENV | BLOCKV | Improve. ↑% |
| LM1B | $2.78 \pm 0.01$ | $2.88 \pm 0.01$ | $3.48 \pm 0.24$ | $2.36 \pm 0.00$ | $2.42 \pm 0.01$ | $2.51 \pm 0.22$ |
| GPT Prompt | $2.88 \pm 0.01$ | $3.00 \pm 0.00$ | $4.33 \pm 0.25$ | $2.43 \pm 0.01$ | $2.51 \pm 0.00$ | $3.43 \pm 0.24$ |
| WebQA | $2.91 \pm 0.01$ | $2.99 \pm 0.01$ | $2.83 \pm 0.65$ | $2.45 \pm 0.01$ | $2.50 \pm 0.01$ | $1.94 \pm 0.61$ |
| PIQA | $2.89 \pm 0.00$ | $2.99 \pm 0.01$ | $3.48 \pm 0.21$ | $2.44 \pm 0.00$ | $2.50 \pm 0.01$ | $2.66 \pm 0.20$ |
| ShareGPT | $2.85 \pm 0.01$ | $2.95 \pm 0.00$ | $3.48 \pm 0.19$ | $2.41 \pm 0.01$ | $2.47 \pm 0.00$ | $2.63 \pm 0.17$ |
| XSum | $2.94 \pm 0.01$ | $3.03 \pm 0.01$ | $3.24 \pm 0.51$ | $2.48 \pm 0.01$ | $2.54 \pm 0.01$ | $2.35 \pm 0.48$ |
| GSM8K | $3.12 \pm 0.01$ | $3.21 \pm 0.02$ | $3.06 \pm 0.95$ | $2.62 \pm 0.01$ | $2.68 \pm 0.02$ | $2.19 \pm 0.89$ |
| WMT-DeEn | $2.75 \pm 0.01$ | $2.83 \pm 0.01$ | $2.99 \pm 0.09$ | $2.33 \pm 0.01$ | $2.38 \pm 0.01$ | $2.18 \pm 0.09$ |
| Average | 2.89 | 2.99 | 3.36 | 2.44 | 2.50 | 2.49 |

Table 7: Speedup comparison between token verification (TOKENV) and block verification (BLOCKV) with $\gamma = 6$ and PALM-2-XXS being the draft model. Each statistic is computed using 1000 test prompts from different datasets on various tasks (each run is an average with 3 different random seeds). Numbers after $\pm$ represent standard deviation.

| Dataset | Block efficiency | | | Wall clock time speedup over baseline | | |
|---|---|---|---|---|---|---|
| | TOKENV | BLOCKV | Improve. ↑% | TOKENV | BLOCKV | Improve. ↑% |
| LM1B | $3.08 \pm 0.01$ | $3.27 \pm 0.01$ | $6.42 \pm 0.07$ | $2.32 \pm 0.01$ | $2.43 \pm 0.01$ | $5.00 \pm 0.06$ |
| GPT Prompt | $3.22 \pm 0.01$ | $3.44 \pm 0.02$ | $6.55 \pm 0.83$ | $2.42 \pm 0.00$ | $2.54 \pm 0.02$ | $5.06 \pm 0.77$ |
| WebQA | $3.26 \pm 0.01$ | $3.44 \pm 0.01$ | $5.60 \pm 0.22$ | $2.45 \pm 0.01$ | $2.55 \pm 0.01$ | $4.24 \pm 0.21$ |
| PIQA | $3.22 \pm 0.02$ | $3.43 \pm 0.02$ | $6.36 \pm 0.78$ | $2.42 \pm 0.01$ | $2.54 \pm 0.01$ | $4.92 \pm 0.72$ |
| ShareGPT | $3.18 \pm 0.02$ | $3.37 \pm 0.01$ | $6.13 \pm 0.53$ | $2.39 \pm 0.02$ | $2.50 \pm 0.01$ | $4.74 \pm 0.49$ |
| XSum | $3.29 \pm 0.01$ | $3.48 \pm 0.01$ | $5.91 \pm 0.82$ | $2.47 \pm 0.01$ | $2.58 \pm 0.01$ | $4.47 \pm 0.77$ |
| GSM8K | $3.56 \pm 0.01$ | $3.80 \pm 0.03$ | $6.86 \pm 0.60$ | $2.66 \pm 0.01$ | $2.80 \pm 0.02$ | $5.38 \pm 0.56$ |
| WMT-DeEn | $3.04 \pm 0.01$ | $3.19 \pm 0.01$ | $4.92 \pm 0.29$ | $2.29 \pm 0.01$ | $2.37 \pm 0.01$ | $3.57 \pm 0.27$ |
| Average | 3.23 | 3.43 | 6.10 | 2.43 | 2.54 | 4.67 |

Table 8: Speedup comparison between token verification (TOKENV) and block verification (BLOCKV) with $\gamma = 4$ and PALM-2-XXXS being the draft model. Each statistic is computed using 1000 test prompts from different datasets on various tasks (each run is an average with 3 different random seeds). Numbers after $\pm$ represent standard deviation.

| Dataset | Block efficiency | | | Wall clock time speedup over baseline | | |
|---|---|---|---|---|---|---|
| | TOKENV | BLOCKV | Improve. $\uparrow$ % | TOKENV | BLOCKV | Improve. $\uparrow$ % |
| LM1B | $2.24 \pm 0.00$ | $2.33 \pm 0.01$ | $4.23 \pm 0.44$ | $2.25 \pm 0.00$ | $2.34 \pm 0.01$ | $3.89 \pm 0.41$ |
| GPT Prompt | $2.41 \pm 0.02$ | $2.48 \pm 0.01$ | $2.96 \pm 1.00$ | $2.42 \pm 0.02$ | $2.48 \pm 0.01$ | $2.72 \pm 0.94$ |
| WebQA | $2.38 \pm 0.01$ | $2.45 \pm 0.01$ | $2.87 \pm 0.13$ | $2.39 \pm 0.01$ | $2.45 \pm 0.01$ | $2.63 \pm 0.12$ |
| PIQA | $2.36 \pm 0.01$ | $2.43 \pm 0.01$ | $3.22 \pm 0.37$ | $2.37 \pm 0.01$ | $2.44 \pm 0.01$ | $2.97 \pm 0.35$ |
| ShareGPT | $2.34 \pm 0.00$ | $2.42 \pm 0.01$ | $3.49 \pm 0.12$ | $2.35 \pm 0.00$ | $2.42 \pm 0.01$ | $3.16 \pm 0.12$ |
| XSum | $2.38 \pm 0.01$ | $2.45 \pm 0.01$ | $2.91 \pm 0.63$ | $2.39 \pm 0.01$ | $2.45 \pm 0.01$ | $2.68 \pm 0.60$ |
| GSM8K | $2.51 \pm 0.01$ | $2.58 \pm 0.02$ | $2.99 \pm 0.47$ | $2.51 \pm 0.01$ | $2.58 \pm 0.02$ | $2.74 \pm 0.44$ |
| WMT-DeEn | $2.22 \pm 0.00$ | $2.28 \pm 0.00$ | $2.59 \pm 0.09$ | $2.24 \pm 0.00$ | $2.29 \pm 0.00$ | $2.37 \pm 0.08$ |
| Average | 2.35 | 2.43 | 3.16 | 2.36 | 2.43 | 2.89 |

Table 9: Speedup comparison between token verification (TOKENV) and block verification (BLOCKV) with $\gamma = 6$ and PALM-2-XXXS being the draft model. Each statistic is computed using 1000 test prompts from different datasets on various tasks (each run is an average with 3 different random seeds). Numbers after $\pm$ represent standard deviation.

| Dataset | Block efficiency | | | Wall clock time speedup over baseline | | |
|---|---|---|---|---|---|---|
| | TOKENV | BLOCKV | Improve. $\uparrow$ % | TOKENV | BLOCKV | Improve. $\uparrow$ % |
| LM1B | $2.36 \pm 0.01$ | $2.48 \pm 0.00$ | $4.93 \pm 0.46$ | $2.27 \pm 0.01$ | $2.37 \pm 0.00$ | $4.55 \pm 0.43$ |
| GPT Prompt | $2.58 \pm 0.04$ | $2.72 \pm 0.02$ | $5.57 \pm 1.29$ | $2.46 \pm 0.03$ | $2.59 \pm 0.01$ | $5.10 \pm 1.22$ |
| WebQA | $2.54 \pm 0.00$ | $2.68 \pm 0.02$ | $5.46 \pm 0.50$ | $2.43 \pm 0.00$ | $2.55 \pm 0.01$ | $5.02 \pm 0.47$ |
| PIQA | $2.50 \pm 0.00$ | $2.62 \pm 0.01$ | $5.06 \pm 0.39$ | $2.39 \pm 0.00$ | $2.50 \pm 0.01$ | $4.66 \pm 0.37$ |
| ShareGPT | $2.47 \pm 0.01$ | $2.60 \pm 0.01$ | $5.10 \pm 0.49$ | $2.37 \pm 0.01$ | $2.48 \pm 0.01$ | $4.69 \pm 0.46$ |
| XSum | $2.54 \pm 0.01$ | $2.67 \pm 0.01$ | $4.83 \pm 0.47$ | $2.43 \pm 0.01$ | $2.54 \pm 0.01$ | $4.45 \pm 0.44$ |
| GSM8K | $2.71 \pm 0.03$ | $2.83 \pm 0.00$ | $4.27 \pm 0.89$ | $2.58 \pm 0.02$ | $2.69 \pm 0.00$ | $3.92 \pm 0.84$ |
| WMT-DeEn | $2.31 \pm 0.01$ | $2.43 \pm 0.02$ | $5.38 \pm 0.57$ | $2.21 \pm 0.00$ | $2.32 \pm 0.01$ | $4.99 \pm 0.54$ |
| Average | 2.50 | 2.63 | 5.07 | 2.39 | 2.50 | 4.67 |

Table 10: Speedup comparison between token verification (TOKENV) and block verification (BLOCKV) with $\gamma = 8$ and PALM-2-XXXS being the draft model. Each statistic is computed using 1000 test prompts from different datasets on various tasks (each run is an average with 3 different random seeds). Numbers after $\pm$ represent standard deviation.

| Dataset | Block efficiency | | | Wall clock time speedup over baseline | | |
|---|---|---|---|---|---|---|
| | TOKENV | BLOCKV | Improve. $\uparrow$ % | TOKENV | BLOCKV | Improve. $\uparrow$ % |
| LM1B | $2.40 \pm 0.01$ | $2.55 \pm 0.01$ | $6.19 \pm 0.43$ | $2.13 \pm 0.01$ | $2.25 \pm 0.01$ | $5.28 \pm 0.40$ |
| GPT Prompt | $2.66 \pm 0.01$ | $2.82 \pm 0.02$ | $6.28 \pm 1.01$ | $2.35 \pm 0.01$ | $2.47 \pm 0.02$ | $5.37 \pm 0.95$ |
| WebQA | $2.61 \pm 0.01$ | $2.78 \pm 0.00$ | $6.27 \pm 0.49$ | $2.31 \pm 0.01$ | $2.43 \pm 0.00$ | $5.39 \pm 0.46$ |
| PIQA | $2.57 \pm 0.01$ | $2.76 \pm 0.01$ | $7.48 \pm 0.51$ | $2.27 \pm 0.01$ | $2.42 \pm 0.01$ | $6.51 \pm 0.47$ |
| ShareGPT | $2.54 \pm 0.01$ | $2.71 \pm 0.01$ | $6.63 \pm 0.72$ | $2.25 \pm 0.01$ | $2.38 \pm 0.01$ | $5.68 \pm 0.68$ |
| XSum | $2.60 \pm 0.01$ | $2.77 \pm 0.00$ | $6.46 \pm 0.49$ | $2.30 \pm 0.01$ | $2.43 \pm 0.00$ | $5.53 \pm 0.46$ |
| GSM8K | $2.82 \pm 0.02$ | $2.98 \pm 0.03$ | $5.48 \pm 1.18$ | $2.49 \pm 0.01$ | $2.60 \pm 0.03$ | $4.62 \pm 1.11$ |
| WMT-DeEn | $2.37 \pm 0.00$ | $2.49 \pm 0.01$ | $5.33 \pm 0.46$ | $2.10 \pm 0.00$ | $2.20 \pm 0.01$ | $4.53 \pm 0.43$ |
| Average | 2.57 | 2.73 | 6.27 | 2.28 | 2.40 | 5.36 |

