# OpenReview forum: "Block Verification Accelerates Speculative Decoding"
_ICLR.cc/2025/Conference — ICLR 2025 Poster_

### Official Review · Reviewer_CsEe · 2024-10-16

**Soundness:** 3
**Presentation:** 3
**Contribution:** 3
**Rating:** 6
**Confidence:** 4

**Summary:**

This paper studies the problem of optimizing speculative decoding from an algorithm level to achieve inference acceleration. The key idea behind this paper is the observation that verifies multiple draft tokens in a block jointly, rather than token-by-token, can bring improvement in mean accepted tokens, which leads to the wall-clock speedup for LLM inference. Based on the idea, the paper proposes a simple block verification algorithm, which is plug-and-play for some existing speculative decoding methods. The authors theoretically show that block verification is lossless and better than standard speculative decoding, making it a stronger default implementation of speculative decoding. The experiments demonstrate that block verification modestly but consistently improves the mean accepted tokens and the final wall-clock speedup.

**Strengths:**

1. The introduction of the block verification in the full distribution is clear and easy to understand, and the paper is well-written with clear explanations to the block verification algorithm.
2. This method shows great simplicity and is easy to use. It does not incur additional computation or memory overhead, and is very easy to implement.
3. The authors theoretically show that block verification is lossless and better than standard speculative decoding, making block verification a stronger default verification for speculative decoding.

**Weaknesses:**

While I really appreciate the simplicity of the method, the novelty and the contributions of the paper are a little weak for me. The reasons are:

1. The motivation that *"verifying multiple draft tokens token by token is not optimal"* is very similar to the motivation in Tree Monte Carlo (TMC) [1], which alleviates the novelty of this paper. Given the empirical results that block verification only brings 5%~8% speedup, I doubt the technical contributions of block verification. The block verification seems more like a trick with theoretical support.
2. There lacks a significant discussion about the decoding temperature $T$ in LLM inference, and the authors do not report the temperature in the experiment details. **In my understanding, the block verification can only bring speedup with $T > 0$, as the block verification degenerates to the token verification in greedy decoding ($T=0$).** Please correct me if I have any misunderstanding. While speculative decoding provides maximal speedup in greedy decoding settings for some specific tasks, such as coding and math reasoning, block verification cannot provide improvement on these tasks. Besides, additional experiments to investigate the influence of different decoding temperature are necessary.
3. Currently, I think there exists a large room (more than 1 page) to conduct more experiments.
   - Experiments for more LLM combinations (e.g. Llama family [2] and Vicuna [3]) across more benchmarks (e.g. MT-Bench [4] and SpecBench [5]) **with different temperature settings**.
   - Experiments for combining block verification to some latest speculative decoding methods (e.g. Medusa [6] and Eagle [7]).

I know that conducting a wide range of evaluation experiments are costly and time consuming, and I will increase my score if the authors clearly address my concerns.



[1] Hu, Zhengmian, and Heng Huang. "Accelerated Speculative Sampling Based on Tree Monte Carlo." *Forty-first International Conference on Machine Learning*.

[2] Dubey, Abhimanyu, et al. "The llama 3 herd of models." *arXiv preprint arXiv:2407.21783* (2024).

[3] Chiang, Wei-Lin, et al. "Vicuna: An open-source chatbot impressing gpt-4 with 90%* chatgpt quality, March 2023." *URL https://lmsys. org/blog/2023-03-30-vicuna* 3.5 (2023).

[4] Zheng, Lianmin, et al. "Judging llm-as-a-judge with mt-bench and chatbot arena." *Advances in Neural Information Processing Systems* 36 (2023): 46595-46623.

[5] Xia, Heming, et al. "Unlocking efficiency in large language model inference: A comprehensive survey of speculative decoding." *arXiv preprint arXiv:2401.07851* (2024).

[6] Cai, Tianle, et al. "Medusa: Simple llm inference acceleration framework with multiple decoding heads." *arXiv preprint arXiv:2401.10774* (2024).

[7] Li, Yuhui, et al. "Eagle: Speculative sampling requires rethinking feature uncertainty." *arXiv preprint arXiv:2401.15077* (2024).

**Questions:**

I would like to raise some questions to improve the manuscripts.

1. Could you provide a more detailed explanation to the motivating example with partial information? You have mentioned "with block verification" in Line 124, but it is not clear how block verification works in this section.
2. Existing speculative decoding methods mainly adopt a tree-based verification manner, which can significantly enhance the mean accepted tokens. Could you provide some explanation how block verification works in a tree-based verification manner?
3. Could you provide a more detailed theoretical assumptions for Definition 1, Theorem 1 and Theorem 2? Besides, I think the section Theoretical Guarantees can be further improved by providing some intuitive explanation (e.g. why block verification is optimal for any valid draft verification algorithm), as this can help readers to understand the advantages of block verification over TMC.

---

> ### Author Response · Authors · 2024-11-22
>
> Thank the reviewer for the detailed reading and helpful comments. Below we address the reviewer's questions.
>
> ***Q1: Comparison to tree Monte Carlo***
>
> The independent work of tree Monte Carlo (Hu and Huang, 2024) shares a similar intuition with our algorithm that more tokens can be accepted by coordinating the acceptance rule of multiple draft tokens. We have proved in our work that our algorithm achieves the optimal speedup among all valid verification algorithms, including theirs (see Theorem 2). We note that Hu and Huang, (2024) does not provide a similar optimality guarantee for their algorithm. Moreover, our algorithm requires minimal changes to the standard token verification algorithm, making it easy to implement in practice. Given that Hu and Huang (2024) did not release their implementation in the paper, it was not possible for us to make an empirical comparison.
>
> ***Q2: Given the empirical results that block verification only brings 5%~8% speedup, I doubt the technical contributions of block verification. The block verification seems more like a trick with theoretical support.***
>
> While the wall clock speedup improvement over standard speculative decoding is modest, the improvement is obtained only through changing the verification algorithm with a few lines of code change and negligible additional computational overhead. In the single-draft case, the standard for this in the literature is the well-established token verification algorithm (Leviathan et al., 2023), which is adopted by almost all follow-up works of speculative decoding in the single-draft case while the drafting mechanism has been improved upon in many works. See Section 6 for a summary of prior works.
> In fact, we show that our proposed algorithm obtained the provably optimal speedup for all valid verification algorithms under natural assumptions (Theorem 2). Hence while the improvement is modest, no other algorithm can do better by just changing the draft verification algorithm, and our algorithm addresses this optimality gap.
> We believe the simplicity, wide applicability and strong theoretical guarantee of the algorithm’s optimality makes this a significant theoretical and algorithmic contribution.
>
> ***Q3: There lacks a significant discussion about the decoding temperature T in LLM inference, and the authors do not report the temperature in the experiment details. In my understanding, the block verification can only bring speedup with T>0, as the block verification degenerates to the token verification in greedy decoding (T=0). ...additional experiments to investigate the influence of different decoding temperatures are necessary.***
>
> The reviewer is correct that block verification is the same as token verification for greedy decoding and no further speedup could be provided. However, we want to emphasize that most modern uses of language models uses temperature sampling with temperature with a positive temperature (see https://community.openai.com/t/cheat-sheet-mastering-temperature-and-top-p-in-chatgpt-api/172683). To study the impact of  temperature, we have added further experiments to numerically quantify the additional speed-up from block verification for temperatures in [0.2, 0.6, 1.0]. Our algorithm also achieves consistent speedups over token verification in these settings. See Section 6.1 in the updated draft and the global response for more detailed results.

---

> > ### Author Response · Authors · 2024-11-22
> >
> > ***Q4: Currently, I think there exists a large room (more than 1 page) to conduct more experiments. (1) Experiments for more LLM combinations (e.g. Llama family [2] and Vicuna [3]) across more benchmarks (e.g. MT-Bench [4] and SpecBench [5]) with different temperature settings. (2) Experiments for combining block verification to some latest speculative decoding methods (e.g. Medusa [6] and Eagle [7]).***
> >
> > Thanks for the suggestions. Regarding *experiments with more LLM combinations*, benchmarks with different temperature settings, we have conducted the set of experiments in SpecBench [5] which includes MT-Bench [4] using the Vicuna [3] family with temperatures in [0.2, 0.6, 1.0]. We observe consistent speedups in these settings as well. See Section 6.1 in the updated draft and the global response for more detailed results and discussions on the impact of temperature.
> >
> > Regarding *multi-draft speculative decoding methods* like Medusa or Eagle, we would like to mention that the submission is mainly focused on the one-draft case. Combining block verification with tree-based verification is an interesting question for future work.
> >
> > While block verification is only designed for single-draft speculative decoding we have already compared with SpecTr and SpecInfer (see Appendix D.1) as two representative multi-draft speculative methods where we observed that for a relatively large batch size, block verification is more effective than these methods. This is important since in practical LLM serving systems, batching is a common technique to increase the throughput of the system.
> >
> > Moreover, the one-draft case is also an active research area. Notably, this is the case that have been studied by a lot of follow-up works including Chen et al [2023b], Sun et al [2024], Zhou et al [2023], Liu et al [2023], Gloeckle et al. [2024], Zhang et al. [2024], Elhoushi et al. [2024] (see a survey of recent of works in Section 6 and Table 2). These works all use TokenVerify as the draft verification algorithm.
> >
> > ***Q5: Could you provide a more detailed explanation to the motivating example with partial information? You have mentioned "with block verification" in Line 124, but it is not clear how block verification works in this section.***
> >
> > Sorry for the confusion. The paragraph following line 124 describes a simplified procedure of the block verification algorithm stated in Algorithm 2. To see this, it can be calculated using the recursive formula in Line 4 of Algorithm 2 (Line 171) and Equation (5), $h_2 = p_2 = 1$ when $X_1X_2 = AB$ or $BB$. Hence $X_1X_2$ are always accepted. When $X_1X_2 = AA$, $h_2 = p_2 = 1/4$, $h_1 = 0$, and $p_{res}(\cdot) = \delta(X = B)$ ($\delta$ represents a point mass). Hence it either accepts both $X_1$ and $X_2$ with probability 1/2 or rejects both tokens and output $B$ from the residual. When $X_1X_2 = BA$, $h_2 = p_2 = 1/2$, $h_1 = 1$, and $p_{res}(\cdot \mid B) = \delta(X = B)$. Hence it accepts both tokens with probability $1/2$, and if not, it always accepts the first token and corrects the second token to $B$.
> > Note that as stated in Algorithm 2, the above quantities can all be calculated based on the **sample path** of the draft block, namely $M_b(\cdot \mid X^{i}), M_s(\cdot \mid X^{i})$, hence it works in the partial information setting.
> > We presented the simplified form and didn’t write out these calculations in Section 2 since the general form of block verification algorithm is not discussed yet, and writing them out might introduce unnecessary new notations, which is not the main purpose of Section 2.
> > We have added the discussions on how the simplified algorithm relates to the general block verification and why it is a particle information verification algorithm in Section 2 of the updated draft (changes are marked in red). Hope the above resolves the reviewer’s question.
> > We are happy to make further modifications if the reviewers have recommendations on how to make the algorithm clearer.

---

> > > ### Author Response · Authors · 2024-11-22
> > >
> > > ***Q6: Existing speculative decoding methods mainly adopt a tree-based verification manner, which can significantly enhance the mean accepted tokens. Could you provide some explanation how block verification works in a tree-based verification manner?***
> > >
> > > Thanks for the insightful question. While tree-based verification for multi-draft speculative decoding is an active area of research, this submission is mainly focused on the one-draft case. Combining block verification with tree-based verification is definitely an interesting question for future work.
> > >
> > > **[Similar to the response to Q4]** While block verification is only designed for single-draft speculative decoding, we have already compared the algorithm with SpecTr and SpecInfer (see Appendix D.1) as two representative multi-draft speculative methods where we observed that for a relatively large batch size, block verification is more effective than these methods. This is important since in practical LLM serving systems, batching is a common technique to increase the throughput of the system.
> > > Moreover, the one-draft case is also an active research area. Notably, this is the case that have been studied by a lot of follow-up works including Chen et al [2023b], Sun et al [2024], Zhou et al [2023], Liu et al [2023], Gloeckle et al. [2024], Zhang et al. [2024], Elhoushi et al. [2024] (see a survey of recent of works in Section 6 and Table 2). These works all use TokenVerify as the draft verification algorithm.
> > >
> > > ***Q7: Could you provide a more detailed theoretical assumptions for Definition 1, Theorem 1 and Theorem 2? Besides, I think the section Theoretical Guarantees can be further improved by providing some intuitive explanation (e.g. why block verification is optimal for any valid draft verification algorithm), as this can help readers to understand the advantages of block verification over TMC.***
> > >
> > > Thanks for the great suggestion. We have updated the Theoretical Guarantees section with more detailed explanations on the theorem and intuitions on how the theoretical guarantees are obtained. See the updated draft for details. The changes are marked in red.

---

> > > > ### Comment · Reviewer_CsEe · 2024-11-22
> > > >
> > > > The rebuttal has addressed most of my concerns, and I appreciate the simplicity of this work. I recommend the authors to contribute the implementation to the community. I will update my score accordingly.

---

### Official Review · Reviewer_jhoE · 2024-11-04

**Soundness:** 3
**Presentation:** 3
**Contribution:** 3
**Rating:** 8
**Confidence:** 3

**Summary:**

The paper introduces a novel verification algorithm called the block verification method, which enhances the efficiency of speculative decoding in large language models (LLMs). This method contrasts with the traditional Token Verification, which optimizes the verification of token blocks to increase efficiency. The study demonstrates that Block Verification outperforms existing methods regarding the expected number of generated tokens (called the block efficiency) and wall-clock times while providing a theoretical guarantee of optimal performance under given conditions. Empirical evaluations show improvements across various tasks, confirming its effectiveness without additional code complexity.

**Strengths:**

- The paper provides solid theoretical guarantees, ensuring that the proposed method does not compromise the output distribution.
- The experiments validate the practical benefits, with performance gains across a diverse set of tasks.
- The proposed algorithm does not increase code complexity, making it easy to integrate into existing LLM architectures.
- The paper highlights how Block Verification compares favorably to related methods, strengthening its contribution.

**Weaknesses:**

- The depth of proofs might be challenging for practitioners without a strong mathematical background, potentially limiting the paper's audience.
- There is a typo in the manuscript: page 8, line 413: deocding -> decoding

**Questions:**

- Are there specific conditions or types of LLM architectures where the performance gains of Block Verification may be more pronounced or limited?
- How does the algorithm handle edge cases where the distributions of the drafting and target models diverge significantly?

---

> ### Author Response · Authors · 2024-11-22
>
> Thank the reviewer for the acknowledging the novelty and strength of our work. Below we answer the reviewers' questions.
>
> ***Q1: The depth of proofs might be challenging for practitioners without a strong mathematical background, potentially limiting the paper's audience.***
>
> Thanks for the useful comment. We have updated the draft with more intuitions on the algorithms parameter choices of the algorithm and how the theoretical guarantees are achieved in Section 4 after stating the theoretical guarantees. The updated discussions are marked in red. We hope this will improve the understandability of the proposed algorithm and its theoretical guarantees.
>
> ***Q2: Are there specific conditions or types of LLM architectures where the performance gains of Block Verification may be more pronounced or limited?***
>
> Our proposed block verification algorithm is independent of the LM architecture and could provide provably optimal speed-up over token verification for any drafter and verifier model. To further demonstrate this, we have added a new set of experiments on Spec-Bench (Xia et al. 2024} with Vicuna family of models implemented in PyTorch. Combined with the original experiments on the PALM-2 family of models implemented in JAX, this demonstrates the advantage of our approach across various LM families, benchmark datasets, and development platforms (see Section 6.1 and the global response for discussions on the result).
> Through experiments, we have identified two factors that affect the speed-up of our proposed method. (1) Draft length, as shown in Figure 5 and the newly added Table 2, the improvement over token verification is higher for longer block length; (2) Temperature: in Table 2, we also conduct various experiments with different temperatures, and we observe that the improvement also increases as the temperature increases.
> The observation is consistent with the intuition behind the algorithm, which coordinates the randomness in the acceptance decisions at different token locations to obtain further improvement. The randomness in the per-token decision increases as the temperature increases and the overall randomness further increases as the draft length increases.
>
> ***Q3: How does the algorithm handle edge cases where the distributions of the drafting and target models diverge significantly?***
>
> When the draft and target model diverge, the verification algorithm will reject the draft tokens with high probability and sample from a residual distribution that is close to the target model. We would like to mention that even when all the tokens are rejected, speculative decoding will still produce a residual token, and hence at least one sample is generated in each decoding iteration; leading to a block efficiency of at least 1.

---

### Official Review · Reviewer_eih8 · 2024-11-04

**Soundness:** 3
**Presentation:** 2
**Contribution:** 3
**Rating:** 6
**Confidence:** 2

**Summary:**

This paper introduces Block Verification, a novel algorithm designed to accelerate speculative decoding in large language models. Unlike traditional token-by-token verification, Block Verification evaluates and accepts a group of tokens (a "block") together. This method achieves faster generation speeds without compromising output quality.

**Strengths:**

1. **Block Verification** improves efficiency by verifying token blocks rather than individual tokens in speculative decoding, preserving the same distribution.

2. Block verification achieves consistent 5–8% speedup and is broadly applicable across models and datasets.

3. Block verification is easy to implement, with minimal modifications to existing speculative decoding systems.

**Weaknesses:**

1. Consistency in final distribution does not guarantee identical generation sequences. For instance, the large model might generate "ABC" while the small model generates "ADC." It’s possible that in the context of "ADC," the next token’s distribution aligns between the models, but under greedy decoding, the large model should ideally follow "ABC" rather than "ADC." The authors state that block verification accepts more tokens than token verification. In the case of greedy decoding, does this imply potential token differences? Would such differences impact the accuracy of the answer, or would block verification revert to token-by-token verification?
2. The explanations of the formulas and algorithms lack intuitive clarity, making comprehension more challenging, even though the modifications from token verification are not large.

**Questions:**

1. In the appendix, a batch size of 8 was used to compare with tree attention. What batch size was used in the main experimental section?
2. How does the performance compare with Medusa/Eagle's static tree attention?

---

> ### Author Response · Authors · 2024-11-22
>
> We thank the reviewer for the helpful comments. Below we answer the reviewers' questions.
>
> ***Q1:  Consistency in final distribution does not guarantee identical generation sequences. For instance, the large model might generate "ABC" while the small model generates "ADC." It’s possible that in the context of "ADC," the next token’s distribution aligns between the models, but under greedy decoding, the large model should ideally follow "ABC" rather than "ADC." The authors state that block verification accepts more tokens than token verification. In the case of greedy decoding, does this imply potential token differences? Would such differences impact the accuracy of the answer, or would block verification revert to token-by-token verification?***
>
> Thanks for the great question. The reviewer is absolutely correct that in the case of greedy decoding, token verification and block verification algorithm will be the same since the distributions will be reduced to a one-hot distribution. Hence, there will be no token differences between the two methods for greedy decoding.  However, we want to emphasize that most modern uses of language models uses temperature sampling with positive temperatures (see discussions in https://community.openai.com/t/cheat-sheet-mastering-temperature-and-top-p-in-chatgpt-api/172683
> ), and we have now added further experiments to numerically quantify the additional speed-up from block verification for temperature in this range (see Section 6.1 and the global response for discussions on the results). Hope this resolves the reviewer’s question.
>
> ***Q2:  The explanations of the formulas and algorithms lack intuitive clarity, making comprehension more challenging, even though the modifications from token verification are not large***
>
> Thanks for the useful comment. We have updated the draft with more intuitions on the algorithms parameter choices of the algorithm and how the theoretical guarantees are achieved in Section 4 after stating the theoretical guarantees. The updated discussions are marked in red. We hope this will improve the understandability of the proposed algorithm and its theoretical guarantees.
>
> ***Q3: In the appendix, a batch size of 8 was used to compare with tree attention. What batch size was used in the main experimental section?***
> Batch size is 1 in all of the main experimental sections. Note that since our method only modifies the verification phase of the algorithm and doesn't introduce additional draft tokens, the speedup we get is independent of the batch size. We have updated the submission to clarify this in the main paper.
>
> ***Q4: How does the performance compare with Medusa/Eagle's static tree attention?***
>
> Medusa and Eagle are multi-draft speculative decoding models. Extending the block verification idea to the multi-draft case like SpecTr and SpecInfer requires non-trivial efforts and is beyond the scope of our work.  Combining block verification with tree-based verification is an interesting question for future work.
> While block verification is only designed for single-draft speculative decoding, we have compared the algorithm with SpecTr and SpecInfer (see Appendix D.1) as two representative multi-draft speculative methods where we observed that for a relatively large batch size, block verification is more effective than these methods. This is important since in practical LLM serving systems, batching is a common technique to increase the throughput of the system.
> Moreover, the one-draft case is an active research area. Notably, this is the case that have been studied by a lot of follow-up works including Chen et al [2023b], Sun et al [2024], Zhou et al [2023], Liu et al [2023], Gloeckle et al. [2024], Zhang et al. [2024], Elhoushi et al. [2024] (see a survey of recent of works in Section 6 and Table 2). These works all use token verification as the draft verification algorithm.
>
> ***Typos.***
> We have fixed the typos in the revised draft.

---

> > ### Comment · Reviewer_eih8 · 2024-11-23
> > **Great Work!**
> >
> > Thank you for your detailed response and thoughtful revisions. Your clarifications and updates have addressed my concerns and further improved the quality of the manuscript. As a result, I have decided to increase my score. Wishing you the best in the next stage of the review process.

---

### Official Review · Reviewer_1Noe · 2024-11-05

**Soundness:** 4
**Presentation:** 3
**Contribution:** 3
**Rating:** 6
**Confidence:** 3

**Summary:**

This paper presents a new algorithm for draft verification for speculative decoding for LLMs. Speculative decoding is a method that accelerates the inference process of LLMs using predictive heuristics to produce “drafts” of the likely next tokens the LLM will generate. These drafts must go through a verification process to determine whether or not they should be accepted, depending on how well they fit the target model’s distribution. The authors argue that the prior state-of-the-art approach for draft verification, Token Verification, is not optimal, since it considers the tokens one by one. The authors instead propose a method to consider the tokens as a block by considering their joint probabilities. The authors also demonstrate that their method accepts more tokens on average than the original Token Verification approach, which is more efficient in the long run. Their experimental results demonstrate that despite the standard overheads of LLM inference, the Block Verification algorithm outperforms the Token Verification algorithm in terms of wall clock speedup.

**Strengths:**

Overall, the paper is well written and the problem is clearly stated.

The theoretical proofs are sound to the best of my knowledge.

**Weaknesses:**

The paper does not provide sufficient evidence that it provides a considerable speedup.
Speedup is from a higher acceptance rate. The paper is using  PALM-2-XXS and XXXS. At this model size, answers tend to be quite bad, which makes it hard for actual response evaluation.

It was not clear how the probabilities for all subblocks are calculated.

# Minor comments

Page 7, line 336: “I.e., it measures” should be “Specifically, it measures”

Page 8, line 425: “all valid verification algorithm” should be “all valid verification algorithms”

Page 8: I think Figure 4 should be labeled as a Table instead of a figure

**Questions:**

- How will the speedup results scale when using larger models?

- How are probabilities of all subblocks calculated?

 - Page 7, line 334: How realistic is this experimental setting? How many drafts are usually produced during speculative decoding, and how many copies of the model would we need for evaluating?

## Post-rebuttal comment

The authors addressed the questions raised in my review, and I have increased my score accordingly.

---

> ### Author Response · Authors · 2024-11-22
>
> We thank the reviewer for the helpful comments and acknowledging the soundedness of our work. Below we answer the reviewer's questions.
>
> ***Q1:  How will the speedup results scale when using larger models? The paper is using PALM-2-XXS and XXXS. At this model size, answers tend to be quite bad, which makes it hard for actual response evaluation.***
>
> We would like to first clarify that the target model (the model where the final generated sequences are following) we use is PALM-2-S. We are using the smaller PALM-2-XXS and XXXS only as fast drafter models, which is typical for speculative decoding implementations.
>
> The number of parameters of the Palm-2-S mode is not publicly known. However, based on the tech report [1], it is a capable model in modern standards. For example, the performance on various benchmarks reported in Table 2 is slightly worse than the previous generation of PALM model with a size of 540B. Additionally, the *one-shot* performance of PALM-2-S reported in [1] exceeds or is on-par with the *multi-shot* performance of Gemma 2 9B model, Mixtral 7B, LLaMA-3 8B (as reported in [2]) for a few benchmarks including TrivialQA (5-shot),  NaturalQuestions (5-shot), HellaSwag (10-shot), and WinoGrande(5-shot).
>
> In the updated draft, we also provide experiments with the open-sourced Vicuna models with Vicuna-7B as the target model and Vicuna 68M as the drafter model, similar to the setting in Spec-Bench [3], as suggested by reviewer CsEe, and we observe similar consistent speedup results (see Section 6.1 in the revised draft and the global response for discussions on the new experimental results).
>
> We hope the above could resolve the reviewer’s concern on the applicability of the method for large models.
>
> ***Q2: How are probabilities of all subblocks calculated?***
>
> After obtaining the next-token probabilities conditioned on the prefixes, the probability of the draft subblocks could be computed by taking the product of each individual draft tokens, e.g., $M(X^k | c) = M(X_1 | c) M(X_2 | c, X_1)... M(X_k | c, X^{k-1})$. In fact, in our implementations of the algorithm, we don’t explicitly compute the probability of these subblocks. Instead, we compute the quantities $p_i$’s as defined in step 4 of Algorithm 2 (line 171), which contains sufficient information the algorithm needs from the probabilities of sub-blocks. A python implementation of the algorithm is described in Appendix A.
>
> ***Q3: Page 7, line 334: How realistic is this experimental setting? How many drafts are usually produced during speculative decoding, and how many copies of the model would we need for evaluating***
>
> As discussed in Leviathan et al., (2023), in practice, the choice of the block length $\gamma$ depends on (1) the improvement in block efficiency when $\gamma$ increases; (2) the increase in drafting time and verification time when $\gamma$ increases. There is usually a sweet spot, which depends on the hardware and similarity of the models. In our experiments, the $\gamma$ that leads to the best speed up is usually 6, and $\gamma = 8$ also achieves a similar speedup. Hence  $\gamma = 8$ is a realistic setting in practice. At the optimal $\gamma = 6$, the algorithm also achieves an additional 6.1% increase in block efficiency and 4.67% in wall clock speedup on average compared to token verification (Leviathan et al., 2023), see Table 7 in the Appendix for the detailed numbers.
>
> Regarding copies of the model during evaluation, if we understand the question clearly, the question is about the number of copies of target model parameters to be loaded into memory during inference. In fact, the parallel evaluation of multiple next-token distributions is implemented through proper setting of the attention mask, and only one copy of the model parameters is loaded during inference.
>
> We hope the above resolves the reviewer’s question. We are happy to answer further questions from the reviewer.
>
> ***Minor comments.***
> We have fixed the typos and updated the title of Figure 4 in the revised draft.
>
> [1] PALM-2 technical report. https://arxiv.org/pdf/2305.10403
>
> [2] Gemma 2: Improving Open Language Models at a Practical Size. https://arxiv.org/pdf/2408.00118
>
> [3] Spec-Bench: A Comprehensive Benchmark and Unified Evaluation Platform for Speculative Decoding https://github.com/hemingkx/Spec-Bench

---

> > ### Comment · Reviewer_1Noe · 2024-11-22
> >
> > Thank you for addressing the questions raised in my review and for additional information in the updated draft.
> > I have increased my score accordingly.

---

### Author Response · Authors · 2024-11-22
**General response and additional experimental results**

We thank all the reviewers for their detailed reading and constructive comments. We have revised the paper accordingly based on the suggestions (the new additions are in red color).
We apologize for the delay in the response since addressing the reviewers’ suggestions required us to implement our algorithm in PyTorch (the original implementations are in JAX) based on codes from Spec-Bench (Xia et al 2024). We are planning to open-source our implementation in a future revision.

Below we discuss *theoretical and practical contributions* of our work, and present the *additional experimental results* on open-sourced models. We will then address each reviewer’s comments in individual responses.

**Theoretical contribution**  We would like to start by highlighting that the main contribution of this work is of  *algorithmic and theoretical* nature. The standard token verification algorithm in speculative decoding proposed by Leviathan et al. (2023) is used in virtually all single draft speculative decoding works (see Section 7 for a summary of related work). We show that this widely adopted algorithm can be improved with theoretical guarantees by proposing block verification. We prove that the algorithm preserves the distribution of the target model (Theorem 1), and it is *optimal* in the expected number of tokens produced in a fixed number of iterations for all possible verification algorithms that guarantee matching output distribution (Theorem 2). Specifically, the expected number of accepted tokens is never worse than the standard token verification algorithm. The theoretical contribution demonstrates the novelty and significance of our work from a theoretical point of view.

**Practical contribution** Practically, our proposed algorithm, block verification, is a *simple* replacement of the standard token verification algorithm in speculative decoding with a few lines of code change, with negligible additional computation overheads, and no changes to the drafting phase. The simplicity of the algorithm makes it compatible with other works that focus on improving the drafting phase of speculative decoding in the single-draft case. Empirically, we have also demonstrated our proposed algorithm consistently achieves modest improvement over standard token verification across a wide range of tasks, models (PALM-2 family and the newly added experiments on open-sourced Vicuna models), temperature ranges (see the newly added experiments), and different mainstream development platforms (PyTorch and JAX), corroborating the theoretical guarantees. While the gains are modest, we believe the simplicity, consistency and strong optimality guarantee makes it a good default implementation in single-draft speculatively decoding.

**New experimental results with open-sourced models** To further demonstrate the effectiveness of our technique in other LLM combinations, especially open-sourced models, benchmarks, and different temperature settings, we conduct the set of experiments proposed in Spec-Bench (Xia et al. 2024) with Vicuna family of models (Chiang et al. 2023}. We use Vincuna-7B-v1.3 as the target model and Vincuna-68M as the draft model. We use a single NVIDIA H100 GPU with a batch size of 1 and a max generation length of 1024. To study the effect of temperature, we consider temperature $T \in \{0.2, 0.6, 1.0\}$ and fix $\gamma = 8$. Our algorithm obtains consistent improvement over token verification  (up to 8.70% in block efficiency (BE) and 6.07% in wall clock speedup (WS) ) across different draft lengths for all temperatures bigger than 0. The results are listed in the table below.  More details about the experiments are added to Section 6.1 in the revised draft.

| $T$  | BE (TokenV) | BE (BlockV) | BE $\uparrow $  | WS (TokenV) | WS (BlockV)| WS $\uparrow \%$  |
| -------- | ------- |------- |------- |------- |------- |------- |
|0.2 | 2.75  | 2.85 | 3.72\% | 1.22  | 1.24 | 1.66\%  |
|0.6 | 2.75  | 2.90 | 5.32\% | 1.23 | 1.29 | 4.24\%  |
|1.0 | 2.79  | 3.04 | 8.70\% | 1.27 | 1.34 | 6.07\%  |

**The effect of temperature**
For temperature 0, which corresponds to greedy decoding, our algorithm is the same as token verification and doesn't provide additional speedups. In non-zero temperature settings, the advantage is consistent, and the additional improvement is higher for larger temperatures. The observation is consistent with the intuition behind the algorithm, which coordinates the randomness in the acceptance decisions at different token locations to obtain further improvement. Modern uses of language models mostly use temperature sampling with temperature with a positive temperature (see https://community.openai.com/t/cheat-sheet-mastering-temperature-and-top-p-in-chatgpt-api/172683
). The result shows that the gain of our algorithm could be higher for more creative generation tasks that use a larger temperature such as creative writing, chatbot response, and exploratory code writing.

---

### Author Response · Authors · 2024-12-04
**Summary of rebuttal and discussion**

We thank the reviewers for their constructive feedback and active engagement in the discussion phase. Based on the reviewers suggestions, we have revised the submission in the following ways:
- Implemented our algorithm in PyTorch and tested it using open-sourced models (Vicuna models) and more public datasets (SpecBench). We will release the implementation in follow-up revisions.
- Included the new experimental results in the revised draft, see Section 6.1 and Table 2.
- Added experiments to study the effect of temperature on the obtained speedup of our method.
- Added more intuitions and discussions to improve the presentation of the algorithm and theoretical results, see e.g., line 285 - line 312 in the revised draft.
- Improved the presentation by fixing typos and adding more explanation in other parts.
See other changes in the draft (highlighted in red).

We'd like to thank the reviewers again for their reviews, which greatly helped improve the submission.

Sincerely,\
Authors

---

### Meta-Review · Area_Chair_jfUh · 2024-12-21

**Metareview:**

(a) Summary of Scientific Claims and Findings

The paper introduces a novel approach to draft verification in speculative decoding that processes multiple tokens as a block rather than individually. The authors demonstrate that this method preserves the target model’s output distribution and achieves optimal performance in terms of tokens accepted per iteration. Empirical results across various datasets indicate a consistent 5%-8% reduction in wall-clock time compared to traditional token-by-token verification, particularly at non-zero decoding temperatures.

(b) Strengths of the Paper

1. Simplicity and Integration: The proposed method is straightforward, requiring minimal changes to existing systems, and integrates easily into speculative decoding workflows.

2. Theoretical Soundness: The approach is backed by rigorous theoretical guarantees, ensuring reliability and optimality.

3. Broad Applicability: The method has demonstrated consistent effectiveness across multiple models (e.g., PALM-2, Vicuna) and development platforms (PyTorch, JAX).

(c) Weaknesses of the Paper and Missing Elements

1. Modest Speedups: The reported 5%-8% improvement in wall-clock time, while consistent, raises questions about the practical significance of the contribution.

2. Limited Experimental Scope: The experiments lack diversity, with no exploration of multi-draft speculative decoding or comparisons to tree-based verification methods.

(d) Decision and Rationale

The method’s simplicity, consistency, and strong theoretical guarantees support its use as a default approach for speculative decoding. However, the modest speedup and limited novelty relative to prior work remain as the weakness of this work.

**Additional Comments On Reviewer Discussion:**

Post-rebuttal revisions clarified explanations and included discussions on the effect of temperature, addressing key reviewer concerns.

The method was recognized as a theoretically optimal and straightforward solution for speculative decoding. Most reviewers adjusted their scores upward following the rebuttal phase.

---

### Decision · Program_Chairs · 2025-01-22

Accept (Poster)